# Semi-Arid-Habitat-Adapted Plant-Growth-Promoting Rhizobacteria Allows Efficient Wheat Growth Promotion

**Nora Saadaoui [1], Allaoua Silini [1], Hafsa Cherif-Silini [1], Ali Chenari Bouket [2], Faizah N. Alenezi [3], Lenka Luptakova [4], Sarah Boulahouat [1] and Lassaad Belbahri [5,*]**

[1]  Laboratory of Applied Microbiology, Department of Microbiology, Faculty of Natural and Life Sciences, Ferhat Abbas University, Setif 19000, Algeria

[2]  East Azarbaijan Agricultural and Natural Resources Research and Education Centre, Plant Protection Research Department, Agricultural Research, Education and Extension Organization (AREEO), Tabriz 5355179854, Iran

[3]  Marine Biodiscovery Centre, Department of Chemistry, University of Aberdeen, Old Aberdeen, Aberdeen AB24 3UE, UK

[4]  Department of Biology and Genetics, Institute of Biology, Zoology and Radiobiology, University of Veterinary Medicine and Pharmacy in Košice, 04181 Kosice, Slovakia

[5]  Laboratory of Soil Biology, University of Neuchatel, 11 Rue Emile Argand, CH-2000 Neuchatel, Switzerland

**\***  Correspondence: lassaad.belbahri@unige.ch

**Abstract:** Plant-growth-promoting rhizobacteria (PGPR) introduced into agricultural ecosystems positively affect agricultural production and constitute an ecological method for sustainable agriculture. The present study demonstrated the effects of two PGPR, *Pantoea agglomerans* strain Pa and *Bacillus thuringiensis* strain B25, on seed germination, on the plant growth of two durum wheat varieties, Bousselam and Boutaleb, and on the frequency of the cultivable beneficial bacterial community. The bacterial strains were used as seed primers (individually or in consortia) by coating them with carboxymethyl cellulose (CMC 1%). The effect of PGPR was negligible on germinative ability but improved seed vigor in the Boutaleb variety after inoculation with the Pa strain alone or in combination with the B25 strain. The results showed that the germination capacity depends on the wheat variety. It seemed to be better in the Bousselam variety. Analysis of the results of morphological plant parameters in sterile compost after 75 days under controlled conditions (16 h light, 26/16 °C day/night) showed a significant improvement in plant growth in both wheat varieties with the Pa strain alone or in combination. Chlorophyll (a, and total), carotenoid, and total soluble sugars were significantly increased, while proline and MDA were strongly reduced by inoculation of the Bousselam variety. Bacterial survival of the Pa and B25 strains in the rhizosphere of sterile compost was appreciable ($10^5$–$10^7$ CFU/g) for both the Pa and B25 strains. Only the Pa strain was endophytic and able to colonize roots. Contrary to sterile compost, the different inoculation treatments in natural soil (after 114 days) significantly improved all morphological parameters and chlorophyll pigments in both wheat varieties. The reduction of proline contents at the leaf level was observed with Pa, mainly in Bousselam. Bacterial densities of the rhizosphere and endophyte cultivable communities did not differ significantly. However, the number of cultivable beneficial bacteria isolated from roots and rhizosphere with multiple PGP traits was significantly increased. Bacterial survival of CMC-coated seed inoculum was appreciable and remained constant, especially for the Pa strain, during 21 months at room temperature. Based on these results, the PGPR used after seed priming would be a feasible and sustainable strategy to improve soil fertility and promote growth of durum wheat in stressful and non-stressful environments.

**Keywords:** PGPR; durum wheat; rhizosphere engineering; germination; plant growth; biopriming

## 1. Introduction

By the middle of the 21st century, the world's population could increase by 26%, adding two billion to the current seven point six billion. Estimates show that to feed nine

point five billion people, agriculture would need to produce 60–70% more grain [1]. United Nations organizations predict continued population growth, and by 2050 this figure is expected to be between 8.3 and 10.9 billion people, while current trends indicate a larger version [2]. Meanwhile, world grain trade in 2017/2018 fell by around 5 million tons (1.2%) to 391 million tons, which was the first reduction in four years [2].

Current agricultural practices, such as the addition of fertilizers, pesticides, and animal waste, have adverse effects on the environment, as they heavily pollute air, soil, and water resources and decrease the biological diversity of ecosystems.

It is therefore necessary to develop new farming strategies [3]. The rising cost of fertilizers and the demand for pesticide-free foods has led to the search for an alternative approach that could alleviate the problem [4]. Modern agriculture offers the potential to sustainably feed the growing world population. So far, genetically modified (GM) products have increased yields and reduced the use of pesticides. Nevertheless, GM products are controversial among policy makers, scientists, and consumers regarding their possible environmental, ecological, and health risks [5].

For the past decades, the response of plant crops to inoculation by plant-growth-promoting rhizobacteria (PGPR) has been studied in numerous experiments carried out around the world in fields and in greenhouses. PGPR are able to exert a beneficial effect on different stages of plant growth. They increase germination percentage, seedling vigor, emergence, root and stem development, total plant biomass, seed weight, early flowering, and fruit and seed yields [6]. Seed vigor and viability are important elements that influence seedling establishment, growth, and crop productivity [7]. Additionally, seed inoculation with PGPR is a promising technology for seed treatment to suppress diseases caused by plant pathogens. PGPR increase germination and seedling vigor at very high rates by reducing the incidence of seed mycoflora [8]. Several studies have mentioned the beneficial effects of bacterial strains on the germination of different plant species under optimal conditions, but especially under stress conditions [9].

PGPR are considered a component for maintaining adequate plant nutrition. PGPR may promote nutrient uptake, thereby reducing the need for fertilizer application and preventing the accumulation of nitrates and phosphates in agricultural soils [10]. The beneficial effect of inoculation on the microbial population may be direct, through an increased supply of available phosphate (P) and nitrogen (N), or indirect, through changes in the growth rate and metabolic activities of crops [11]. Based on the data obtained, it is evident that inoculation resulted in significant increases in the yields of different crops, under different conditions [12]. Treatments with PGPR can affect the growth and yield of a wide range of crops such as cereals or legumes [13,14]. A large number of bacteria, such as *Bacillus* spp., *Azospirillum* spp., *Pseudomonas* spp., *Azotobacter* spp., *Pantoea* spp., and others, have significantly improved wheat yield parameters [15–18].

Understanding the integration of bacterial strains in the rhizosphere and the mechanisms of their interactions is a key element in improving and stimulating plant growth. Microbiomes associated with plant roots have attracted particular attention in recent years [19,20]. Numerous studies have shown that plant-associated microbiota play an important role in plant growth and development and are able to provide plant protection against pathogens and various abiotic stresses [21,22]. The plant microbiome supplies the host plant with additional gene pools and is therefore often referred to as the second plant genome [23].

Understanding the fate of this microbiota is fundamental to developing smart farming practices. Rhizospheric microbiomes are regulated by root exudates, plant genotype selection, and environmental adoption. Thus, the plant microbiome must have an ability to interact with beneficial microbes from natural soil microbiota or microbial inoculants because the agronomic response to inoculation also depends on native microbial communities [24]. The question is whether inoculation can lead to changes in community structure by increasing the plant-growth-promoting and disease-suppressing functions of the resident community [25]. A growing number of researchers are recognizing the effects

of exogenous inoculants, not only on promoting plant root growth, but also on protecting roots from disease damage by regulating the rhizosphere microbiota from the seedling stage. Plants have important regulatory effects on the formation of the microbial structure of the rhizosphere by excreting different root exudates at different growth stages [26]. Plants have the ability to modify the soil environment by secreting bio-active molecules into the rhizosphere to alter the edaphic conditions of the soil microbiota [27]. Thus, different plant species or genotypes can recruit specific microbiota through differences in root morphology and root exudation patterns [21]. In addition, the composition of root exudates and the structure of the microbial community associated with the roots are strongly affected by the growth stage of the plant [28]. An active early-stage rhizosphere microbial community is an important stage of community diversity and succession during the growth period of the whole plant [29,30]. Then, inoculation with *Azospirillum lipoferum* CRT1 affected the size and taxonomic composition of the functional communities involved in nitrogen fixation and ACC deamination [31]. Similarly, inoculation of barley seeds (*Hordeum vulgare*) by *Bacillus* increased the total number of bacteria and the population of phosphate-solubilizing bacteria [14]. Bio-inoculants by PGPR or arbuscular mycorrhizal fungi (AMF) induced a significant modification of the structure of the wheat bacterial community [32]. Soybean inoculation with *Paenibacillus mucilaginosus* decreased bacterial richness and diversity but also favored specific micro-organisms through selectivity and enrichment of root exudates. Additionally, many bacterial classes and genera, which were associated with symbiotic nitrogen fixation, plant growth promotion, biological control, and soil activity enhancement, were overrepresented [33]. Besides growth parameters, defense enzymes, soil enzymes, and microbial diversity were significantly modulated in plants inoculated individually and in consortia with *Pseudomonas putida* NBRIRA and *Bacillus amyloliquefaciens* NBRISN13 [34].

The present study was designed to evaluate the role of PGPR on the improvement of morphological and biochemical parameters of durum wheat at the germination and plant-growth stages. Two durum wheat varieties were used by bio-priming the seeds with a carboxymethylcellulose (CMC) coating. These PGPR *Pantoea agglomerans* strain Pa and *Bacillus thuringiensis* strain B25 were used as seed primers in two common strategies (single strains and consortium). These PGPR were tested at the germination stage, then under axenic conditions (sterile compost) to explore their ability to survive in the rhizosphere, and finally in natural soil to assess their impact on plant growth and the beneficial bacterial community of wheat. Further, these PGPR were also tested for their ability to survive on the coated seeds during storage.

## 2. Materials and Methods

### 2.1. Bacterial Strains

The bacterial strains *Pantoea agglomerans* Pa (LMA2) (MUJJ00000000.1) [18] and *Bacillus thuringiensis* B25 (JX196352) [35] were used in this study. Strain Pa was originally isolated from the rhizosphere of durum wheat fields in the arid and saline region of Bou-Saâda, Algeria (35°23′38″ N 4°19′18.1″ E, pH 9, electrical conductivity (EC = 3.54 ms/cm). Strain B25 was isolated from the rhizosphere of durum wheat in the semi-arid region north of Sétif, Algeria (36°17′40.5″ N 25°32.8″ E, pH 7.9, EC = 1.3 ms/cm). The strains were selected as the best bacterial isolates with high PGP activities and tested for none antagonism between them, allowing to use them either individually or in combination in order to synergistically promote plant growth.

### 2.2. Plant Material

The seeds of two local durum wheat genotypes, Bousselam (*Triticum durum* L.c.v Bousselam) and Boutaleb (*Triticum durum* L.c.v Boutaleb), were used. Their area of adaptation were the high plateaus and the interior plains of eastern Algeria. Bousselam variety (Pedigree: Heider/Marli/Heider-Cro ICD-414-1BLCTR-4AP) was a semi-late variety, with medium height. Boutaleb variety (Pedigree: HEDBA03/OFONTO–DZ–ITGC-SET001-) has an intermediate vegetative cycle. The grain and straw yields were high in both wheat

varieties, and they presented good tolerance to abiotic stresses (lodging, drought, and cold) and biotic stresses (fungal diseases). They were kindly provided by the Technical Institute of Field Crops (Sétif, Algeria).

### 2.3. Effect of Bacterial Inoculation on Seed Germination

The experiment was performed to evaluate the effects of the PGPR on seed germination, which is considered the first step of plant growth.

### 2.3.1. Seed Disinfection

The seeds were disinfected on the surface by successive immersion in ethanol (70%, 1 min), then in sodium hypochlorite (2%, 30 min) and rinsed several times with sterile distilled water [18].

### 2.3.2. Preparation of Bacterial Inoculum and Seed Coating

The bacterial strains were taken from the glycerol stock and inoculated onto nutrient agar at 30° C/24 h. Each strain was inoculated in trypticase soy broth (TSB) with constant shaking at 150 rpm for 48 h/30 °C. The cultures were centrifuged at 3000 rpm for 20 min. A pellet of each strain was suspended in 1% sterile carboxymethyl cellulose (CMC) solution at $10^8$ CFU/mL. For preparation of microbial consortia, the mixture of bacterial strains was prepared by mixing individual bacterial inocula at equal volume to form a suspension containing the cells of each strain. The sterilized seeds were then carefully coated with homogenized CMC containing bacterial cells, shaken for 3 h/30 °C, air-dried overnight under aseptic conditions in a laminar air flow, and stored at room temperature in the dark. Un-inoculated seeds coated in sterile CMC solution served as a negative control [18].

### 2.3.3. Seed Germination

Germination was carried out on bacterized and coated seeds in triplicate, with four groups representing the type of treatment:

- T1: Un-inoculated seeds (Control)
- T2: Seeds inoculated with Pa
- T3: Seeds inoculated with B25
- T4: Seeds co-inoculated with Pa + B25

Thirty seeds from each treatment were placed in Petri dishes on a double layer of filter paper soaked with 10 mL of sterile distilled water. The plates were incubated in the dark at 25 ± 2 °C. The germinated seeds were counted every three days (3, 6, and 9 days). When the radicle was at least 3 mm long, the seeds were considered to be germinated. The incubation was maintained over 11 days. Four parameters were recorded from this experiment: final germination percentage (FGP), germination rate index (GRI), seedling length vigor index (SLVI), and seedling weight vigor index (SWVI) and were calculated as described by Kerbab et al. [14].

$$FGP = \text{Number of seeds germinated/Total number of seeds} \times 100$$

$$GRI = G3/3 + G6/6 + G9/9 \text{ (G3, G6, and G9 were germination percentages at 3, 6, and 9 days)}$$

$$SLVI = \text{Seedling length (cm)} \times (\%) \text{ germination}$$

$$SWVI = \text{Seedling dry weight (mg)} \times (\%) \text{ germination}$$

### 2.4. Effect of Bacterial Inoculation on Plant Growth of Durum Wheat in Sterile Compost

To evaluate the effect of inoculation on plant growth in sterile compost, the wheat seeds of Bousselam and Boutaleb varieties, sterilized as previously, were germinated on filter paper in Petri dishes containing 10 mL of sterile distilled water at 20 °C/48 h in the dark. When radicle length reached 3 mm, inoculation and coating in CMC solution were performed as described before. Un-inoculated seeds (control) were immersed in a 1% sterile

CMC solution. Plastic pots ($\varnothing$ = 10 cm) disinfected with a sodium hypochlorite solution were filled with 150 g of compost sterilized at 120 °C/1 h for 3 successive days. The pots were divided into four groups representing the type of treatment as previously. The treated seeds were sown (one seed/pot) at a depth of 1 cm from the surface. The experiment was repeated 4 times and was conducted for 75 days in a growth chamber with an average day/night temperature of 26 °C and 16 °C, respectively, and 16 h light photoperiod. The humidity of the compost was adjusted and kept constant by watering with sterile water. At the ear stage, roots and shoots were collected. Their lengths and fresh and dry weight were determined. Dosages of biochemical parameters of growth (chlorophyll pigments and total sugars) and stress (proline and malondialdehyde: MDA) were carried out. The survival of the inoculated bacteria in the rhizosphere and their capacity for root colonization were evaluated.

### 2.4.1. Measurement of Morphological Parameters

After 75 days of growth, the plants were harvested and washed with distilled water. Roots and shoots were separated. Lengths of shoots and roots (cm) and fresh and dry weight (after 72 h at 65 °C) of shoots and roots (g) were measured. Morphological measurements were performed four times.

### 2.4.2. Measurement of Biochemical Parameters
### Chlorophylls and Carotenoids

Chlorophylls a and b and carotenoids were performed according to Kerbab et al. [14]. An amount of 0.5 g of the leaves of each sample was cut into small segments (0.5 cm), homogenized in 10 mL of 80% acetone, and stored at −10 °C overnight. The organic extract was centrifuged at 14,000 rpm/5 min and the absorbance of the supernatant was measured by spectrophotometer at OD 663, 645, and 470 nm to determine chlorophylls a ($Chl_a$), b ($Chl_b$), and carotenoids, respectively.

$$Chl_a = 12.70\ A_{663} - 2.69\ A_{645}$$

$$Chl_b = 22.90\ A_{645} - 4.68\ A_{663}$$

$$Chl_{a+b} = 20.21\ A_{645} + 8.02\ A_{663}$$

$$Carotenoids = (1000\ A_{470} - 1.9\ Chl_a - 63.14\ Chl_b)/214$$

### Total Sugars

The extraction of total sugars from the fresh leaf material was carried out according to the following protocol: 3 mL of ethanol (80%) was added to 0.1 g of leaves. The mixture was incubated at room temperature in the dark for 48 h. Then, the mixture was heated at 80 °C in a water bath to evaporate the ethanol, then 20 mL of distilled water was added. Total sugars were determined according to Dubois et al.'s [36] method. The reaction mixture had contained 0.5 mL of the sample, 0.5 mL of a phenol solution (5%), and 2.5 mL of concentrated sulfuric acid. The color intensity proportional to sugars concentration was measured by spectrophotometer at OD 490 nm. The values obtained were translated into glucose concentrations by reference to a previously established calibration curve. All biochemical parameters were performed in duplicate.

### Proline

Leaf proline extraction was performed by cold mixing 50 mg aliquots of fresh leaf weight with 1 mL of ethanol:water (40:60 *v/v*) solution and left overnight at 4 °C. The mixture was then centrifuged at 14,000 rpm/5 min. An amount of 500 μL of the supernatant was added to 1000 μL of the reaction mixture (1% (*w/v*) ninhydrin in 60% (*v/v*) acetic acid) in 1.5 mL screw cap tubes and heated at 95 °C/20 min. The mixture was then centrifuged at 10,000 rpm and the evaluation of the leaf proline content was determined at OD 520 nm using proline (5 mM) as a standard solution [18].

Lipid Peroxidation (MDA)

Lipid peroxidation in leaves was determined by estimation of malondialdehyde (MDA) content. Indeed, 0.2 g of the fresh material was cut into 5 mm pieces and macerated in 1 mL of trichloroacetic acid (0.1%) and centrifuged (10,000 rpm/5 min). An amount of 0.5 mL of the supernatant was added to 2 mL of TCA (20%) containing 0.5% thiobarbituric acid. The mixture was heated at 95 °C/30 min and then rapidly cooled in an ice bath. The mixture was centrifuged at 10,000 rpm/15 min and the absorbance of the supernatant was measured at OD 532 nm. the OD was corrected to remove non-specific turbidity by subtracting the OD 600 nm. The MDA concentration was calculated using the extinction coefficient of 155 mM$^{-1}$cm$^{-1}$ [37].

### 2.4.3. Bacterial Survival in the Rhizosphere

Bacterial persistence in the rhizosphere was determined by counting (CFU/g) cultivable bacteria after 75 days of wheat plant growth. An amount of 1 g of sterile rhizospheric compost taken from the surface of roots was homogenized with 10 mL of sterile physiological water for 5 min. A serial dilution of up to $10^{-6}$ of the samples was spread on the surface of TSA medium and incubated at 30 °C/48 h [18]. The survival of the bacteria expressed in CFU/g of soil was carried out in duplicate.

### 2.4.4. Endophytic Colonization

Endophytic bacteria detection was performed by washing the roots from adhering compost by stirring at 300 rpm/5 min. The roots were disinfected at the surface by immersion in 70% ethanol/1 min, then in 2% sodium hypochlorite/30 min and rinsed several times with sterile distilled water. To check the efficiency of root disinfection, the final washing water was spread on TSA agar and the plates were incubated at 30 °C/48 h. An amount of 1 g of sterilized roots was macerated in 10 mL of sterile physiological water and ground using a mortar. A serial dilution of up to $10^{-6}$ of the samples was spread on the surface of TSA medium and incubated at 30 °C/48 h. Bacterial counts were expressed in CFU/g of roots and carried out in duplicate.

### 2.5. Effect of Inoculation on Durum Wheat Plant Growth in Non-Sterile Soil

The soil was taken from a field located at the University of Sétif 1, Algeria (36°12′00.8″ N 5°22′08.9″ E). The physicochemical analysis was carried out at the Fertial agronomic laboratory, Annaba, Algeria. The soil was classified as clayey and loamy, with electrical conductivity and pH of 0.15 mS/cm and 8.34, respectively. The contents of readily available organic matter, nitrogen, K, and phosphorus were, respectively, 14.1 g/kg, 1.3 g/kg, 183 mg/kg, and 14 mg/kg.

Plastic pots (20 cm in diameter × 18 cm in height), disinfected with a sodium hypochlorite solution, were filled with approximately 4 kg of soil that was previously sieved through a 0.5 cm mesh and air-dried. Wheat seeds were disinfected, germinated, and coated as previously described. Twenty seeds were sown in each pot. The treatments were carried out in five repetitions for each wheat variety, Bousselam and Boutaleb. Five treatments were performed as follows: (1) seeds not inoculated, (2) seeds inoculated with Pa, (3) seeds inoculated with B25, (4) seeds co-inoculated with Pa and B25, and the last treatment, (5) the seeds were inoculated with the bacterium *Bacillus velezensis* FZB 42 [38] used as a positive control.

Plant growth was monitored in a controlled growth chamber (16 h light photoperiod, 26/16 °C day/night temperatures). Plants were thinned 15 days after sowing to maintain the desired uniform number (ten seedlings per pot) in each pot. The plants were irrigated twice a week with potable water. After growing for 114 days (from 28 November 2019 to 21 March 2020), the plants, at the ear stage, were harvested and washed with distilled water. Roots and shoots were separated, and growth parameters (morphological and biochemical) were measured.

### 2.5.1. Determination of Morphological Parameters

Lengths of shoots and roots (cm), and fresh and dry weights (after 72 h at 65 °C) of shoots and roots (g) were measured. Morphological measurements were performed in triplicate.

### 2.5.2. Biochemical Parameters

Biochemical parameters: chlorophylls a and b, carotenoids, total sugars, proline, and MDA contents were determined in duplicate, as described previously.

### 2.6. *Effect of Inoculation on the Rhizobacterial Community*

### 2.6.1. Enumeration of Rhizobacteria

Wheat rhizospheric soil (1 g) adhering tightly to the roots was homogenized in 10 mL of sterile physiological water and shaken for 5 min. An amount of 100 μL of the sample and of each decimal dilution ranging from $10^{-1}$ to $10^{-6}$ was spread out in duplicate on TSA medium and incubated at 30 °C/72 h. Bacterial counts were expressed in CFU/g soil and performed in duplicate. Based on the morphological characteristics of the colonies, 13 visibly distinct isolates were purified and stored in agar slants at 4 °C for further evaluation of their PGP activities.

### 2.6.2. Enumeration of Endophytic Bacteria

The roots were disinfected as described previously. An amount of 1 g of disinfected roots was ground and homogenized in 10 mL of sterile physiological water. A serial dilution of up to $10^{-6}$ of the samples was spread on the surface of a TSA medium and incubated at 30 °C/72 h. Bacterial counts were expressed in CFU/g of roots and performed in duplicate. From six to eight strains were purified and stored for characterization of their PGP activities.

### 2.7. *PGP Activities*

This assay was performed in order to evaluate the impact of the bacterial inoculation on the improvement of PGP traits of rhizospheric and endophytic bacterial communities. The PGP activities of all the isolates were determined in vitro in duplicate, using standard protocols for phosphate solubilization, nitrogen-free growth (nitrogen fixation), siderophore and IAA production, and ACC deaminase activity.

### 2.7.1. Phosphate Solubilization

Phosphate solubilization was determined by culturing the isolates in the Pikovskaya's agar (PVK) medium [39]. An amount of 10 μL of each bacterial culture was spotted on the PVK agar surface and incubated at 30 °C for 7 days. The development of clear zones around the colonies were considered as positive phosphate solubilizers [13].

### 2.7.2. Nitrogen Fixation

Nitrogen fixation was tested on Winogradsky (WS) solid medium without nitrogen. An amount of 10 μL of each bacterial culture was spotted on the agar medium surface. The growth of a visible colony on the medium after 4 days of incubation at 30 °C indicated the ability of the strain to fix nitrogen. The reaction of each bacterial strain was rated positive or negative in the test [40].

### 2.7.3. Production of Siderophores

Bacterial isolates were screened for siderophore production using Chrome Azurol S (CAS) medium [41]. Fresh bacterial culture (100 μL) was transferred into 10 mL of iron-restricted King's B medium and incubated at 30 °C/96 h. The cultures were centrifuged at 12000 rpm/5 min and 100 μL of the supernatant was mixed with 100 μL of CAS reagent in a microplate well and incubated for 30 min in the dark. A change in the blue color of the

medium to orange indicates the production of siderophores. OD 630 nm was measured by a microplate reader and the percentage of siderophore units was estimated using the formula:

$$\text{Decolorization (\%)} = [(A_r - A_s)/A_r] \times 100$$

where $A_r$ is the absorbance of the reference sample and $A_s$ is the absorbance of the sample. The color changes from blue to orange according to the amount of siderophores produced.

### 2.7.4. IAA Production

Indole-3-acetic acid (IAA) production was evaluated by culturing bacterial isolates in Luria–Bertani broth supplemented with tryptophan (2 g/L). Fresh bacterial culture (100 μL) was transferred into 10 mL of LB medium and incubated at 30 °C/96 h. The cultures were centrifuged at 12,000 rpm/5 min, then 100 μL of the supernatant was transferred in a microplate well and 150 μL of Salkowski reagent (50 mL, 35% of perchloric acid, and 1 mL FeCl$_3$ 0.5 M solution) was added. The microplate was incubated for 30 min in the dark. The absorbance was read at 490 nm. IAA concentrations (μg/mL) were determined by a calibration curve of pure IAA [42].

### 2.7.5. ACC Deaminase Production

1-aminocyclopropane-1-carboxylic acid (ACC) deaminase activity of the isolates was evaluated according to Li et al.'s [43] method, using the nitrogen-free DF medium [44]. A bacterial colony was cultured in 5 mL of LB broth incubated at 28 °C/48 h with shaking at 150 rpm. Then, 2 mL of each culture was centrifuged at 8000 rpm/5 min. The cell pellet was recovered, washed twice with 1 mL of DF liquid medium, and suspended in 2 mL of DF-ACC medium (DF medium amended with ACC at 3.0 mmol/L as sole nitrogen source). After incubation at 28 °C/24 h, 1 mL of the bacterial culture was centrifuged and 100 μL of the recovered supernatant was diluted 10 times in liquid DF medium without ACC in 1.5 mL tubes. Then, 60 μL of each supernatant was mixed with 120 μL of ninhydrin reagent (500 mg of ninhydrin and 15 mg of ascorbic acid were dissolved in 60 mL of ethylene glycol, stored at −20 °C, and mixed with 60 mL of citrate buffer (mol/L, pH 6.0)). The reaction mixture was heated by boiling for 30 min. The qualitative test was appreciated with the naked eye by comparison with DF solution without inoculation. A bacterial isolate, which makes a deeper color of supernatant compared with that of the DF-ACC medium without inoculation, was considered as an ACC-utilizing bacterial isolate.

### 2.8. Effect of Storage on Bacterial Survival of Coated Seeds

To evaluate the effects of the shelf life on the bacterial survival of the coated seeds, 1 g of seed, taken at different periods of storage at ambient temperature (24 h, 45 days, and 7, 13, and 21 months), was suspended in 10 mL of physiological water. Bacterial counting was performed in duplicate on TSA surface medium at 30 °C/48 h, as described previously. The bacterial enumeration was expressed in CFU/g of seeds.

### 2.9. Statistical Analysis

The statistical analysis of the data was performed using analysis of variance (ANOVA) and when significant effects were detected, the groups were compared using a post hoc Tukey's HSD test. The level of significance used for all statistical tests was 5% ($p < 0.05$). The statistical program used was IBM SPSS Statistics v.22.

## 3. Results

### 3.1. Effect of Bacterial Inoculation on Seed Germination

The inoculation of two varieties of wheat with the Pa, B25, or Pa + B25 strains did not induced any significant changes in the final percentage and the seed germination index (Figure 1A,B and Figure 2), but it did allow a significant increase in vigor index of length (SLVI) (Figure 1C) and weight (SWVI) (Figure 1D). This was observed only in the Boutaleb

variety, when the seeds were treated with the Pa strain alone or in combination with B25 strain. Concerning the varietal response of wheat to germination, a better germination capacity of inoculated and non-inoculated seeds was observed in the Bousselam variety (Figure 1A,B).

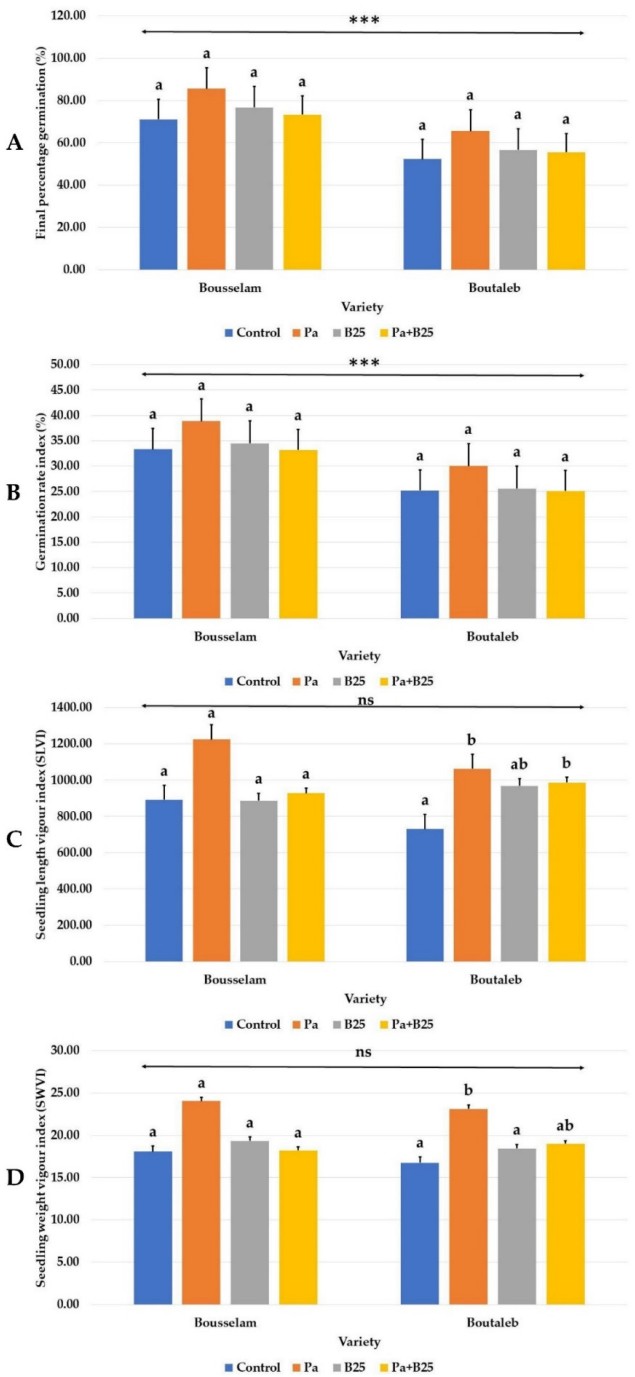

**Figure 1.** Effect of bacterial inoculation on (**A**) final percentage germination (%), (**B**) germination rate index (%), (**C**) seedling length vigor index, and (**D**) seedling weight vigor index of the two wheat varieties, Bousselam and Boutaleb. The bar plots represent the mean ± standard error of three different experiments. IBM SPSS Statistics v.24 was used to perform statistical analysis, using a two-way ANOVA and Tukey's multiple comparison post-test. Significant differences are displayed as: *** $p < 0.001$. ns: non-significant.

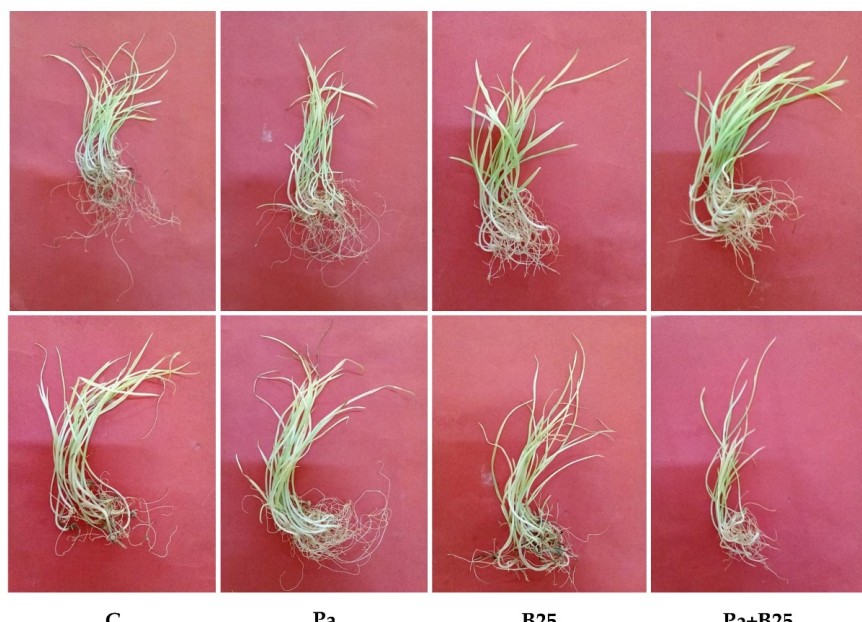

C          Pa          B25          Pa+B25

**Figure 2.** Effect of bacterial inoculation on in vitro wheat seedling germination. **Top**: Bousselam variety. **Bottom**: Boutaleb variety.

### 3.2. Effect of Bacterial Inoculation on Plant Growth of Wheat in Sterile Compost

### 3.2.1. Morphological Parameters

Analysis of the results of fresh and dry weights of roots and leaves, root elongation, and height of the plant showed that bacterial inoculation or co-inoculation led to a significant improvement in all these parameters for the two wheat varieties, with the exception of B25 in Bousselam variety (Figures 3A–F and 4).

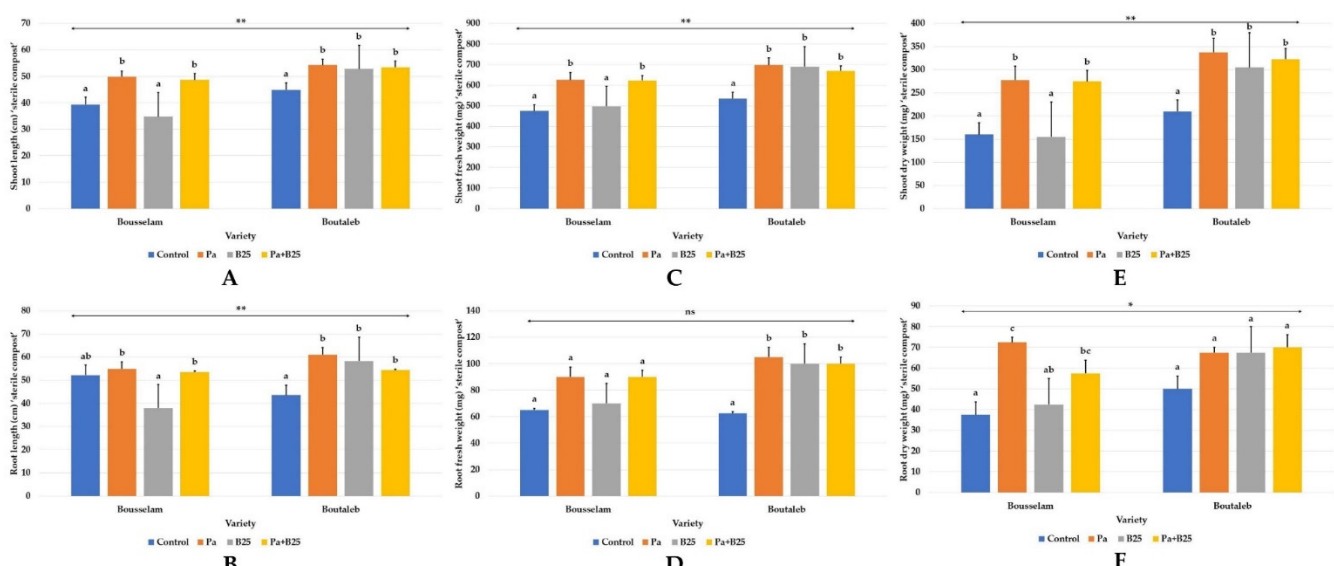

**Figure 3.** Effect of bacterial inoculation on (**A**,**B**) shoot and root length (cm), (**C**,**D**) shoot and root fresh weight (g), and (**E**,**F**) shoot and root dry weight of wheat plants grown in sterile compost of Bousselam and Boutaleb varieties. The bar plots represent the mean ± standard error of three different experiments. IBM SPSS Statistics v.24 was used to perform statistical analysis, using a two-way ANOVA and Tukey's multiple comparison post-test. Significant differences are displayed as: * $p < 0.05$, ** $p < 0.01$. ns: non-significant.

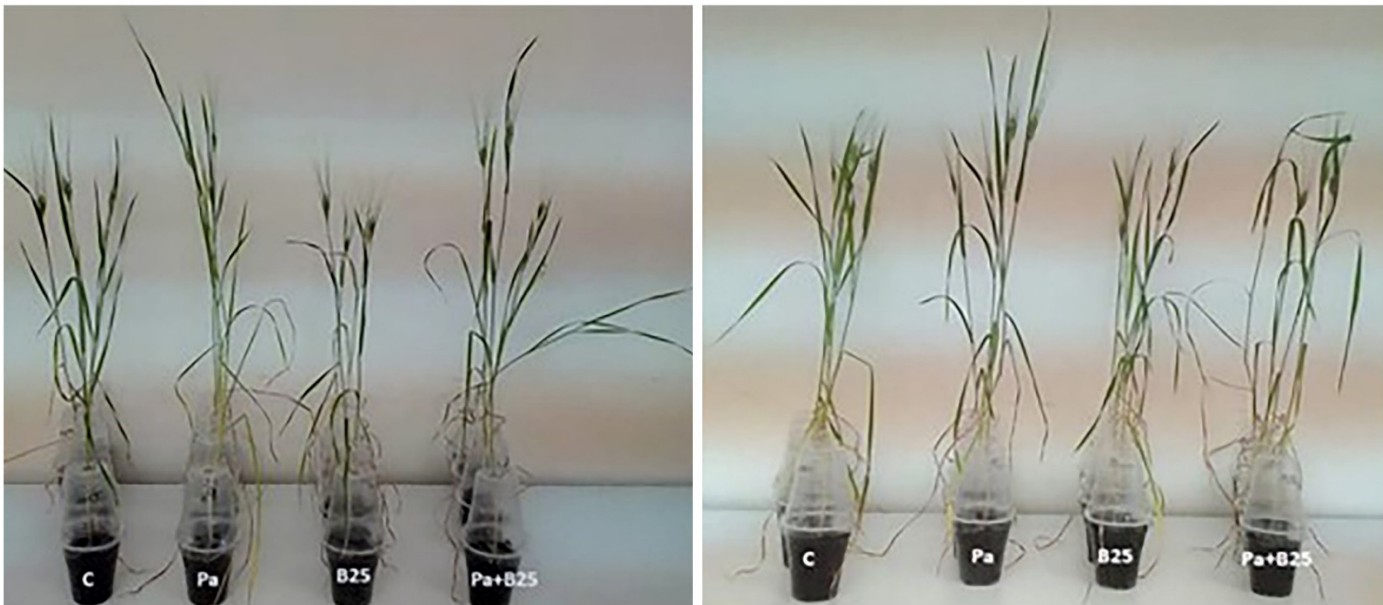

**Figure 4.** Effect of bacterial inoculation on in vitro wheat plants grown in sterile compost. (**Left**) Bousselam variety. (**Right**) Boutaleb variety.

### 3.2.2. Biological Parameters

Chlorophyll Pigments

Compared to the control, the content of chlorophyll pigments (chlorophyll a, b, total, and carotenoids) increased with Pa strain inoculation alone or in co-inoculation with B25 strain, while B25 strain treatment seemed to have a non-significant effect on the content of these pigments (Figure 5). In the Bousselam variety, a significant increase of 83.75% in chlorophyll a and 74.17% in total chlorophyll was recorded in the case of co-inoculation (Figure 5A,C). The content of carotenoids was also improved under this treatment in Bousselam and Boutaleb varieties, with rates of 46.63%, and 74.16%, respectively (Figure 5D). Significant differences between wheat varieties were observed only for chlorophylls b and total.

Total Sugars, Proline, and MDA

The effect of inoculation on leaf total sugar content was variable depending on the wheat variety and was manifested by a significant increase in leaves of plants treated with Pa or B25 strains or with co-inoculation in the Boutaleb variety. In Bousselam variety, only the Pa strain alone or in combination had a positive impact (Figure 6A).

The effects of seed inoculation on biochemical indicators of stress in both wheat varieties appeared to be mixed. Indeed, variations in leaf proline and MDA content following seed inoculation were observed only in the Bousselam variety. Decreases in proline were noticed when the Pa strain in mono-inoculation or co-inoculation with B25 strain were used (Figure 6B) and in MDA during the different treatments (Figure 6C).

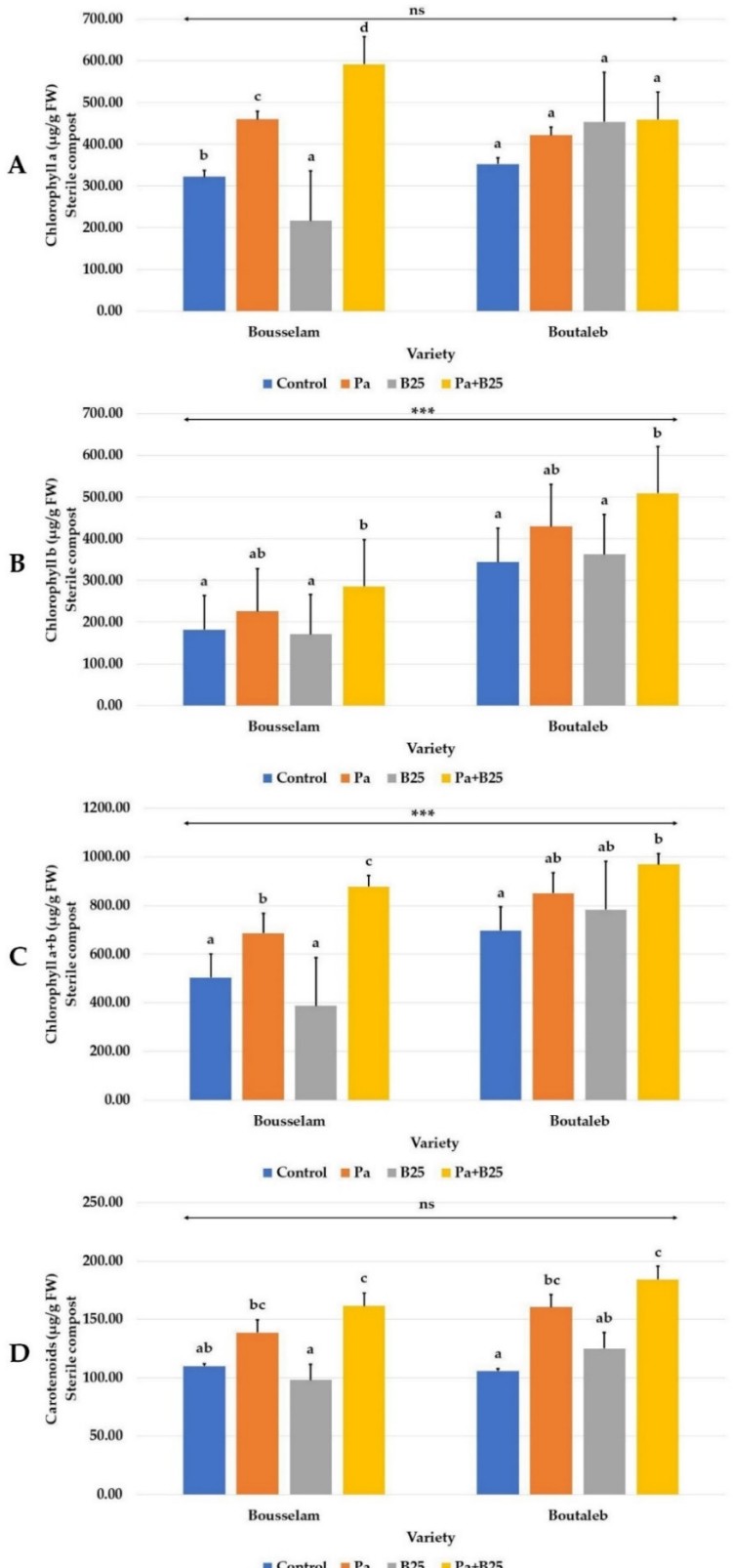

**Figure 5.** Effect of bacterial inoculation on (**A**) chlorophyll a (μg/g FW), (**B**) chlorophyll b (μg/g FW), (**C**) chlorophyll a + b (μg/g FW), and (**D**) carotenoid (μg/g FW) contents of wheat plants grown in sterile compost of Bousselam and Boutaleb varieties. The bar plots represent the mean ± standard error of three different experiments. IBM SPSS Statistics v.24 was used to perform statistical analysis, using a two-way ANOVA and Tukey's multiple comparison post-test. Significant differences are displayed as: *** *p* < 0.001. ns: non-significant.

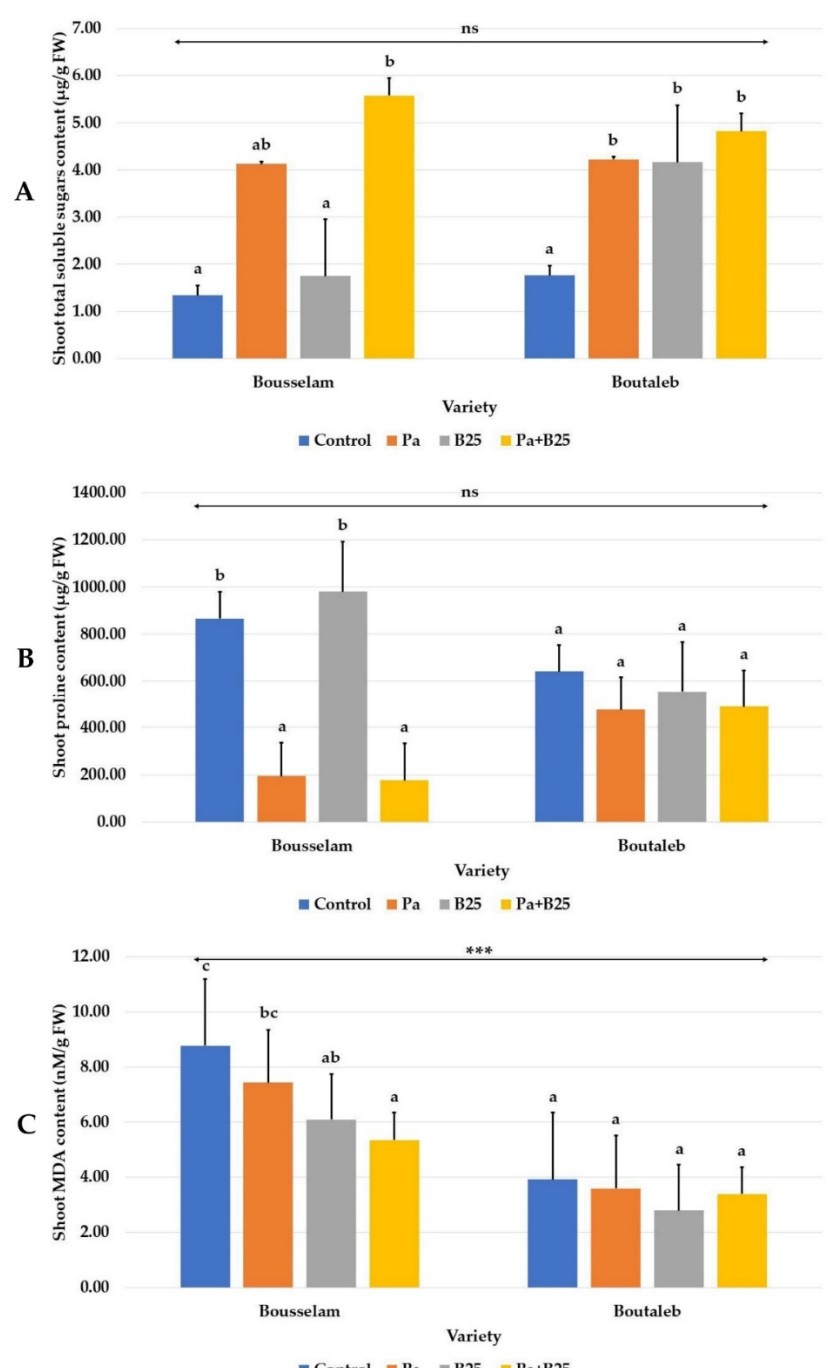

**Figure 6.** Effect of bacterial inoculation on (**A**) shoot total soluble sugars content (mg/g FW), (**B**) shoot proline content (μg/g FW), and (**C**) shoot malondialdehyde (MDA) content (μM/g FW) of wheat plants grown in sterile compost of Bousselam and Boutaleb varieties. The bar plots represent the mean ± standard error of three different experiments. IBM SPSS Statistics v.24 was used to perform statistical analysis, using a two-way ANOVA and Tukey's multiple comparison post-test. Significant differences are displayed as: *** $p < 0.001$. ns: non-significant.

### 3.2.3. Bacterial Survival in Sterile Rhizosphere

After 75 days of plant growth, bacterial survival of the Pa and B25 strains in the rhizosphere of both wheat varieties was appreciable. The enumeration of the Pa strain in mono- and co-inoculation was about $1.8 \times 10^6$ CFU/g and $2.91 \times 10^7$ CFU/g in the rhizosphere of the Bousselam variety, (Figure 7A), and about $1.19 \times 10^7$ CFU/g and $2.30 \times 10^5$ CFU/g in the rhizosphere of the Boutaleb variety, (Figure 8B). Thus, for the B25 strain,

taken individually or in association with the Pa strain, the enumerations were about $2.10 \times 10^5$; $4.73 \times 10^7$ and $7.86 \times 10^6$; $6.91 \times 10^6$ CFU/g in the rhizospheric soil of the Bousselam and Boutaleb varieties, respectively. Inside the roots, only the Pa strain existed at levels of $1.08 \times 10^6$ and $3.50 \times 10^6$ in Bousselam variety roots and $9.33 \times 10^5$ CFU/g and $2.22 \times 10^8$ CFU/g $10^6$ in Boutaleb variety roots. Unlike B25, the Pa strain was able to colonize roots and appeared to be an endophytic bacterium.

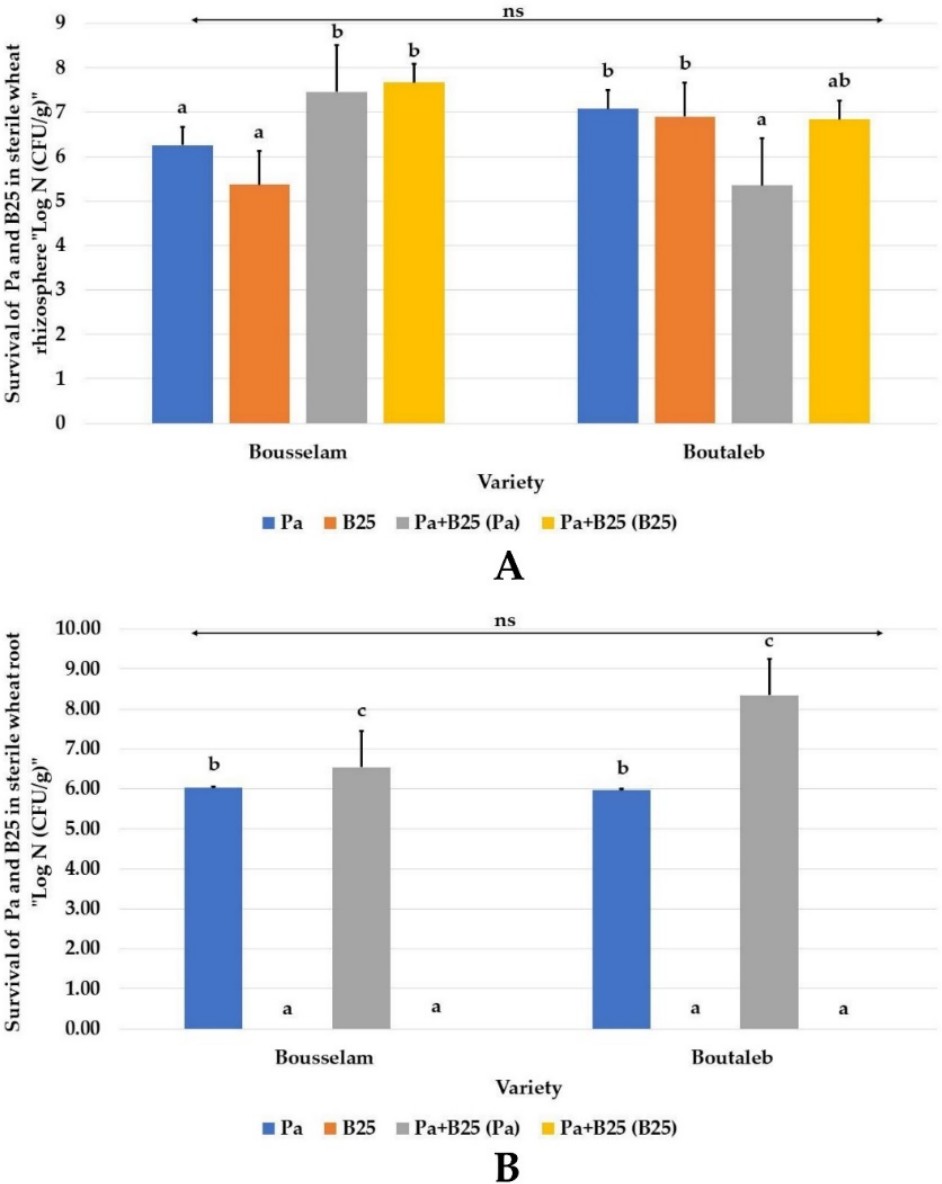

**Figure 7.** Survival of bacteria, Pa and B25 (Log N(CFU/g)), in (**A**) rhizosphere and (**B**) roots of wheat plants of the Bousselam and Boutaleb varieties, grown in sterile compost inoculated with Pa, B25, and Pa + B25. The bar plots represent the mean ± standard error of three different experiments. IBM SPSS Statistics v.24 was used to perform statistical analysis, using a two-way ANOVA and Tukey's multiple comparison post-test. ns: non-significant.

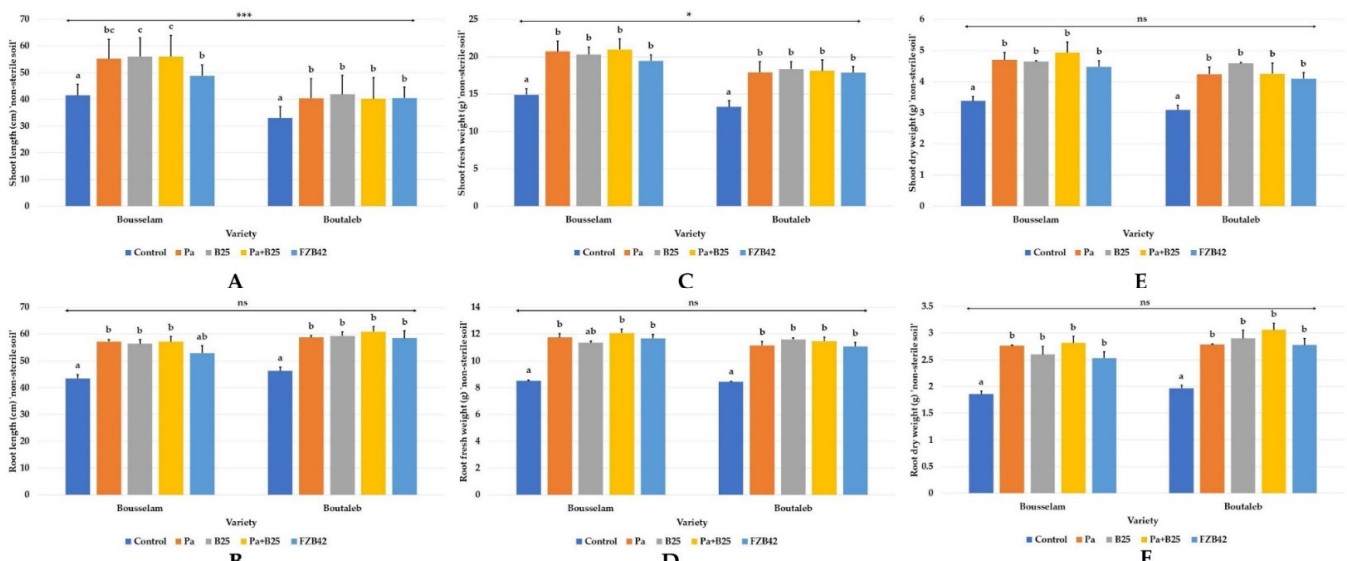

**Figure 8.** Effect of bacterial inoculation on (**A**,**B**) shoot and root length (cm), (**C**,**D**) shoot and root fresh weight (g), and (**E**,**F**) shoot and root dry weight of wheat plants grown on non-sterile soil of Bousselam and Boutaleb varieties. The bar plots represent the mean ± standard error of three different experiments. IBM SPSS Statistics v.24 was used to perform statistical analysis, using a two-way ANOVA and Tukey's multiple comparison post-test. Significant differences are displayed as: * $p < 0.05$, *** $p < 0.001$. ns: non-significant.

*3.3. Effect of Inoculation and Co-Inoculation on Durum Wheat Plant Growth in Non-Sterile Soil*

3.3.1. Morphological Parameters

Compared to sterile compost, in non-sterile soil, inoculation in all treatments significantly improved the morphological growth parameters in both wheat varieties (Figures 8 and 9) but with greater increases in plant height (Figure 8A,B) and root fresh weight (Figure 8C,D) in the Bousselam variety.

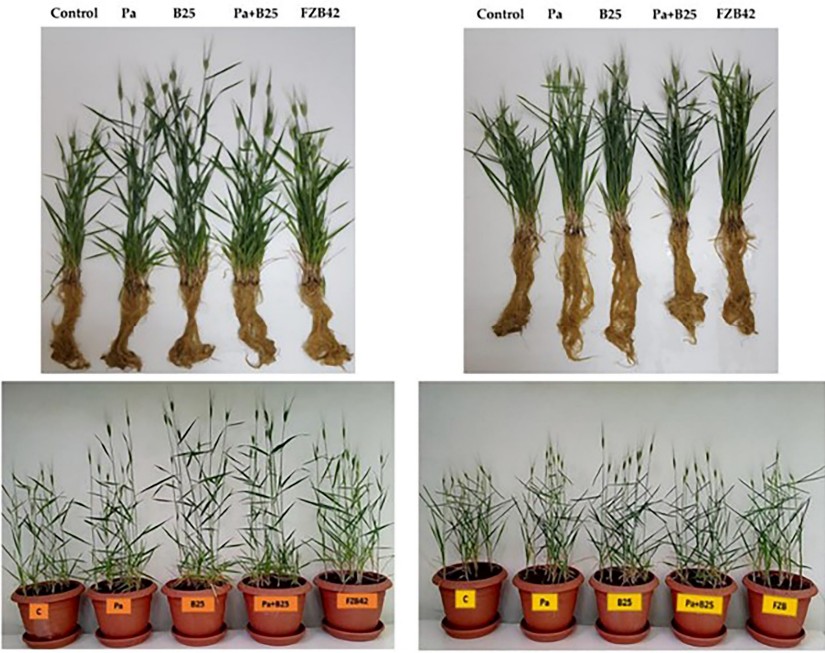

**Figure 9.** Effect of bacterial inoculation on in vitro wheat plants grown on non-sterile soil. (**Left**) Bousselam variety. (**Right**) Boutaleb variety.

### 3.3.2. Biochemical Parameters

Chlorophyll Pigments

Bacterial inoculation resulted in significant increases in chlorophyll pigment levels, but the Pa strain performed best (Figure 10A–D). It should be noted that the effect of the variety on these biochemical parameters was not detectable.

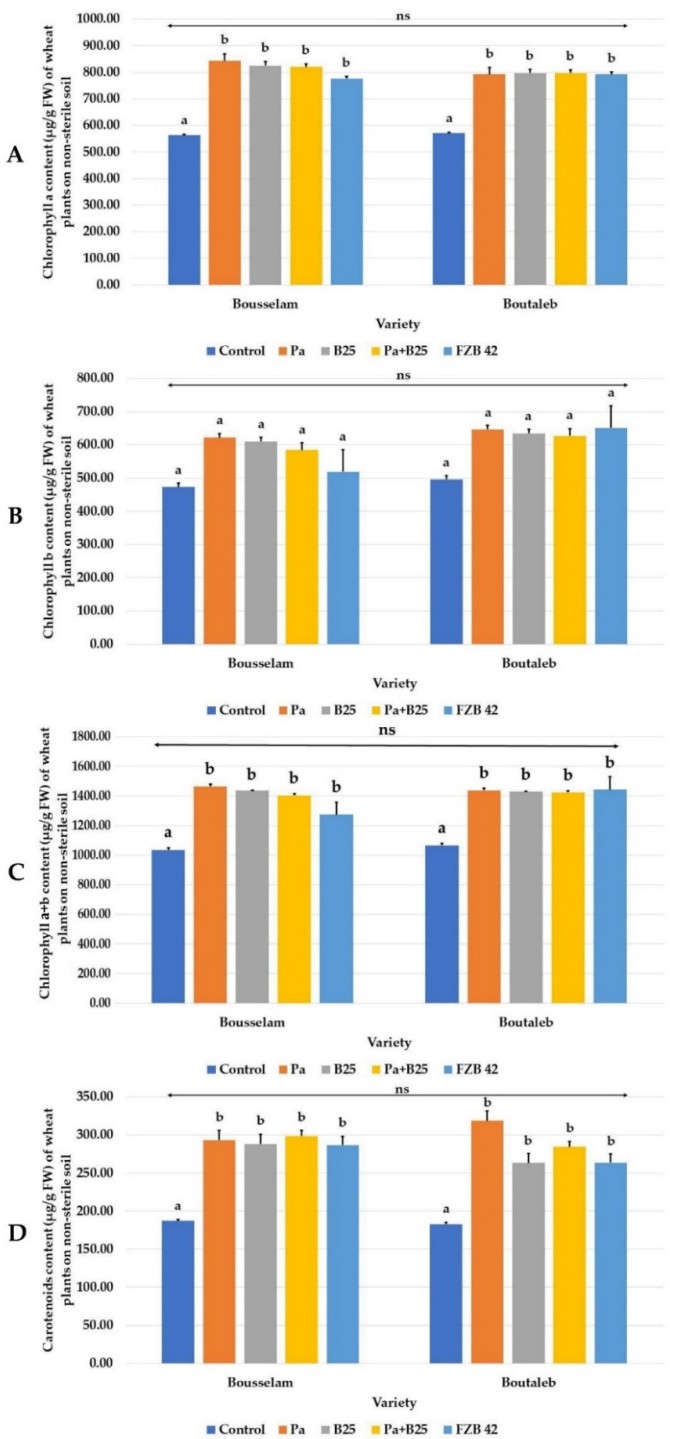

**Figure 10.** Effect of bacterial inoculation on (**A**) chlorophyll a (µg/g FW), (**B**) chlorophyll b (µg/g FW), (**C**) chlorophyll a + b (µg/g FW), and (**D**) carotenoid (µg/g FW) contents of wheat plants grown on non-sterile soil of Bousselam and Boutaleb varieties. The bar plots represent the mean ± standard error of three different experiments. IBM SPSS Statistics v.24 was used to perform statistical analysis, using a two-way ANOVA and Tukey's multiple comparison post-test. ns: non-significant.

Total Sugars, Proline, and MDA

Inoculation had no significant effect on leaf total sugar content in either wheat variety (Figure 11A). Regarding leaf proline content, the results were mixed and variable depending on the wheat variety studied. A significant reduction in the concentration of this amino acid was obtained only with the FZB42 strain in the Bousselam variety, whereas in Boutaleb variety, this reduction was observed following the effect of the different inoculation treatments (Figure 11B). As for the leaf content in MDA, it was variable and depended on the variety used; it remained unchanged in the Boutaleb variety but decreased strongly in the Bousselam variety under the effects of the Pa and B25 strains, alone or in combination (Figure 11C).

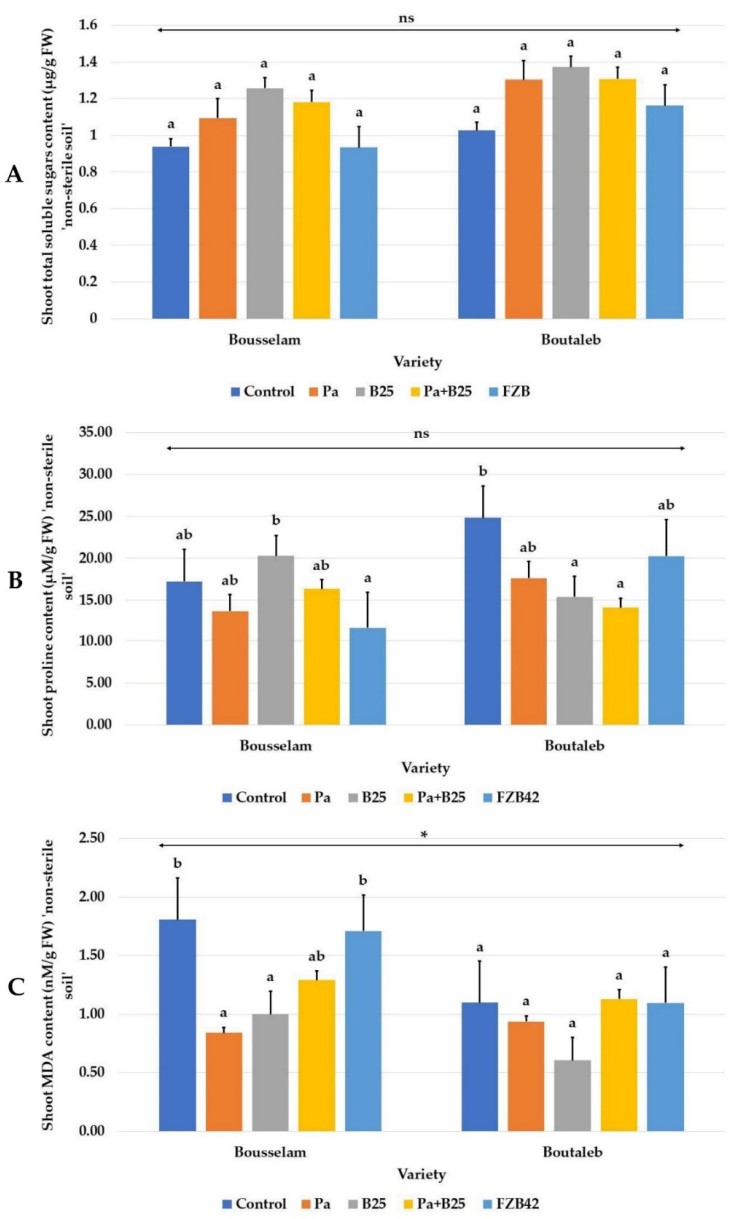

**Figure 11.** Effect of bacterial inoculation on (**A**) shoot total soluble sugars content (mg/g FW), (**B**) shoot proline content (μg/g FW), and (**C**) shoot malondialdehyde (MDA) content (μM/g FW) of wheat plants grown on non-sterile soil of Bousselam and Boutaleb varieties. The bar plots represent the mean ± standard error of three different experiments. IBM SPSS Statistics v.24 was used to perform statistical analysis, using a two-way ANOVA and Tukey's multiple comparison post-test. Significant differences are displayed as: * $p < 0.05$. ns: non-significant.

### 3.3.3. Effect of Inoculation on the Rhizobacterial and Endophyte Community
Total Cultivable Bacterial Community

Evaluation of the total cultivable bacterial density of rhizospheric and endophyte communities of wheat roots under the effect of Pa, B25, FZB42, or Pa + B25 strains compared to that of non-inoculated plants revealed no significant differences in the two wheat varieties (Figure 12). Enumerations of total bacteria were in the range of $10^5$ to $10^6$ CFU/g of soil or root for rhizospheric bacteria and endophytes.

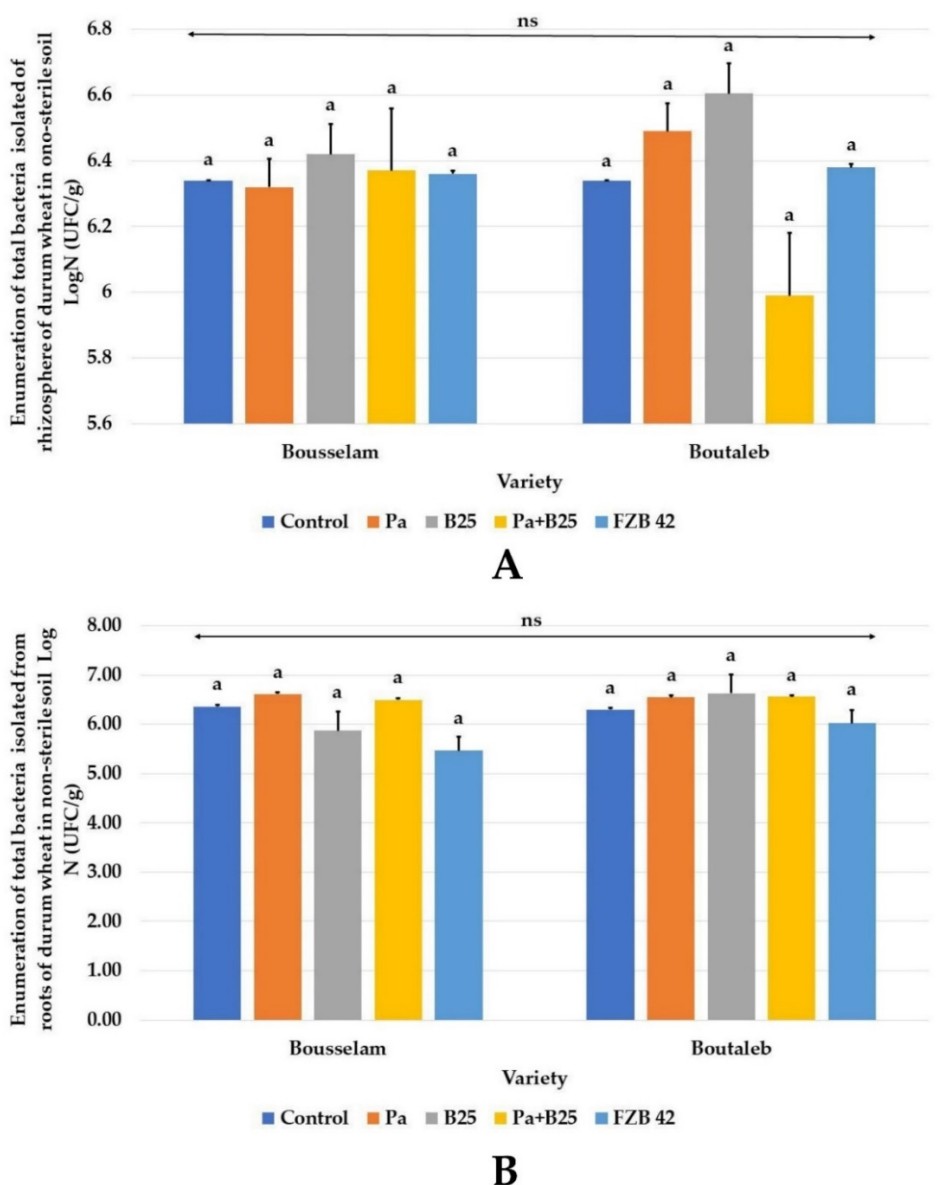

**Figure 12.** Enumeration of total bacteria isolates (Log N(CFU/g)) from (**A**) rhizosphere and (**B**) roots of wheat plants of Bousselam and Boutaleb varieties, grown in non-sterile soil inoculated with Pa, B25, Pa + B25 and FZB42. The bar plots represent the mean ± standard error of three different experiments. IBM SPSS Statistics v.24 was used to perform statistical analysis, using a two-way ANOVA and Tukey's multiple comparison post-test. ns: non-significant.

### 3.3.4. Cultivable PGPR Community

A set of 40 endophytic isolates from the roots of the Bousselam variety revealed that, among them, 15 isolates possessed all PGP activities—namely $N_2$ fixation, IAA, siderophore, and ACC deaminase production. These isolates came from plants inoculated with PGPRs,

namely B25 (n = 5), Pa (n = 4), and Pa + B25 (n = 3), FZB42 (n = 3). Only three isolates were endophytes from non-inoculated plants (Figure 13A). In the Boutaleb variety, a total of 30 isolates, including 6 isolates per treatment, were evaluated for their PGP activities. Thirteen isolates had all the PGP abilities, including five strains isolated from Pa + B25 inoculated roots, three from Pa and FZB42 inoculated roots, and only one strain isolated from B25 inoculated roots and non-inoculated roots (Figure 13B).

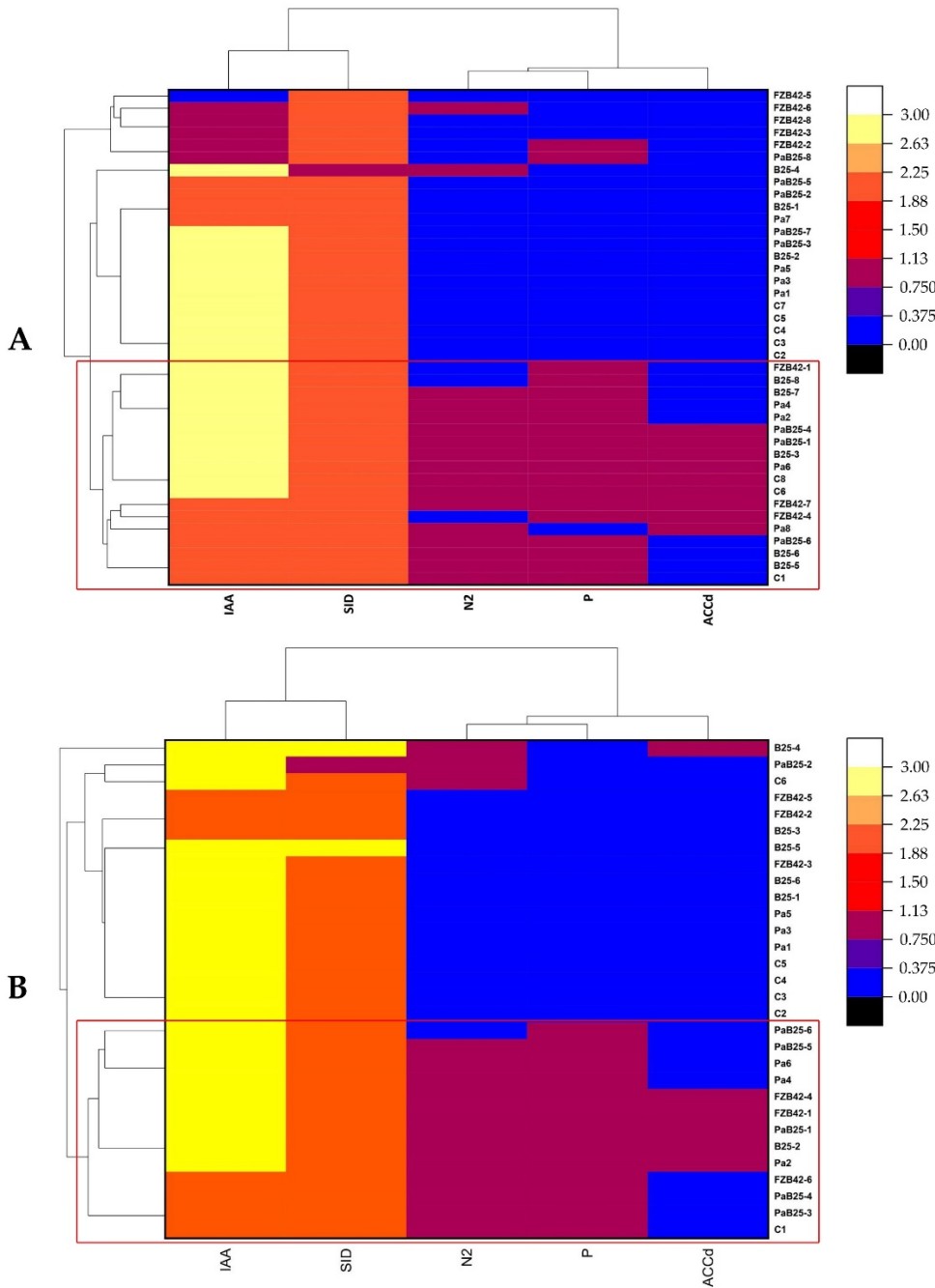

**Figure 13.** Clusters and heat map of endophytic bacteria from the durum wheat roots inoculated with Pa, B25, FZB42, and Pa + B25 and the expression of their PGP activities (IAA: indole acetic acid production, SID: siderophore production, $N_2$: nitrogen fixation, P: phosphate solubilization, and ACCd: 1-aminocyclopropane-1-carboxylic acid deaminase production). (**A**) Bousselam variety; (**B**) Boutaleb variety.

In contrast to the endophytic isolates from the roots of the two wheat varieties, Bousselam and Boutaleb, the number of rhizospheric isolates was greater (n = 13) for each treatment. Strains expressing the best PGP activities came from plants inoculated with Pa, B25, and FZB42, or co-inoculated with Pa + B25, including 17 strains from the Bousselam variety and 35 from the Boutaleb variety (Figure 14A,B).

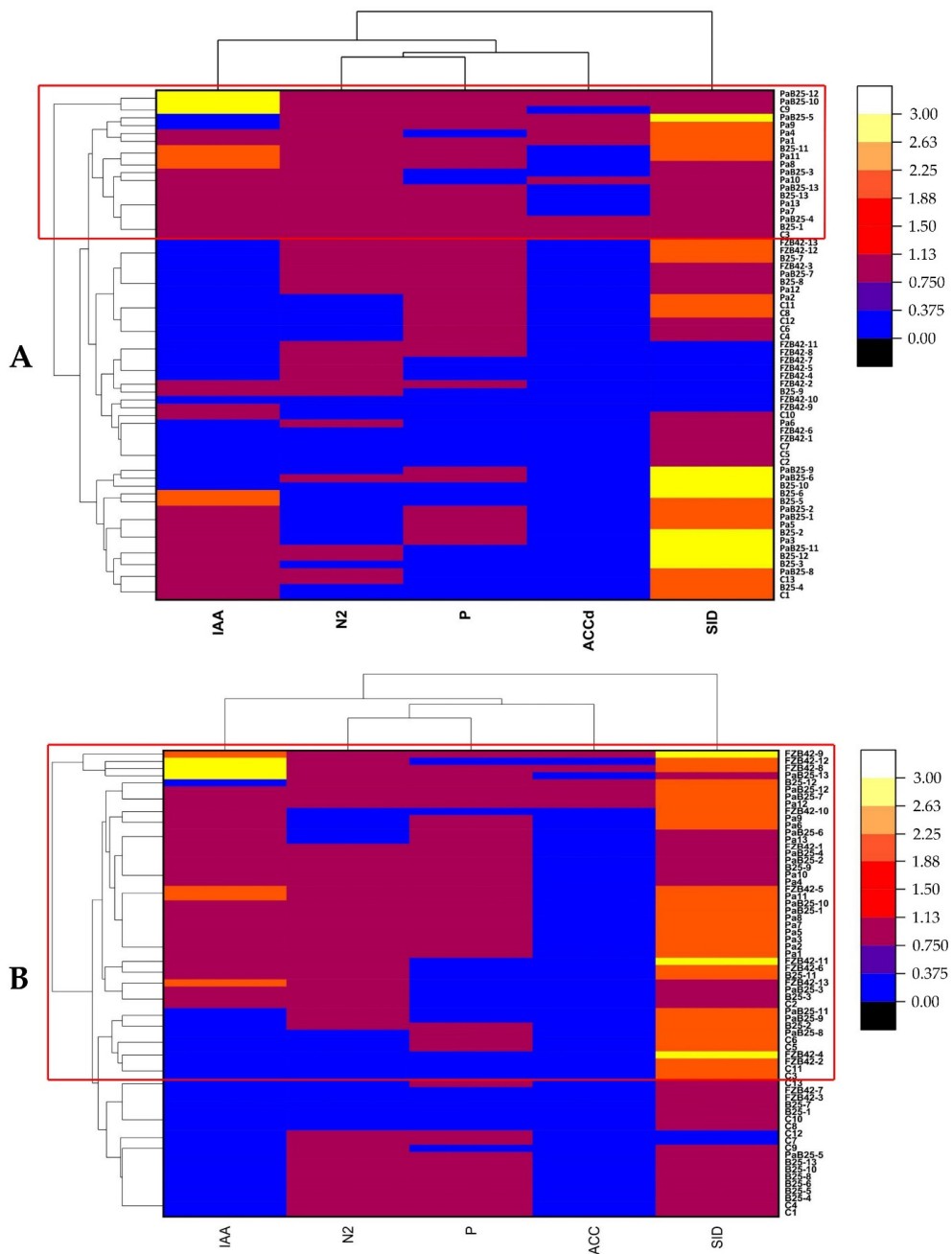

**Figure 14.** Clusters and heat map of bacteria isolated from rhizosphere of durum wheat inoculated with Pa, B25, FZB42, and Pa + B25 and the expression of their PGP activities (IAA: indole acetic acid production, SID: siderophore production, N2: nitrogen fixation, P: phosphate solubilization, and ACCd: 1-aminocyclopropane-1-carboxylic acid deaminase production). (**A**) Bousselam variety; (**B**) Boutaleb variety.

### 3.4. Effect of Storage on Bacterial Survival of Coated Inoculated Seeds

In both wheat varieties, the initial bacterial number of the two strains encapsulated with carboxymethyl cellulose (CMC), either individually or in consortium, was in the

range of $10^6$ to $10^8$ CFU/g of seeds. The Pa strain appeared to be the more stable during seed storage at room temperature in both cases. Indeed, in the Bousselam variety, the enumerations of the Pa strain at T = 0 (0 days) were $1.56 \times 10^7$ CFU/g in the mono-inoculation, and $2.24 \times 10^7$ CFU/g in the co-inoculation. After 21 months, they reached $6.05 \times 10^5$ and $3.80 \times 10^5$ CFU/g, respectively (Figure 15). On seeds of the Boutaleb variety, the initial number was $2.92 \times 10^8$ CFU/g in mono-inoculation, and $2.79 \times 10^7$ CFU/g in consortium. After 21 months, this number reached $6.70 \times 10^7$ and $3 \times 10^5$ CFU/g, consecutively (Figure 15).

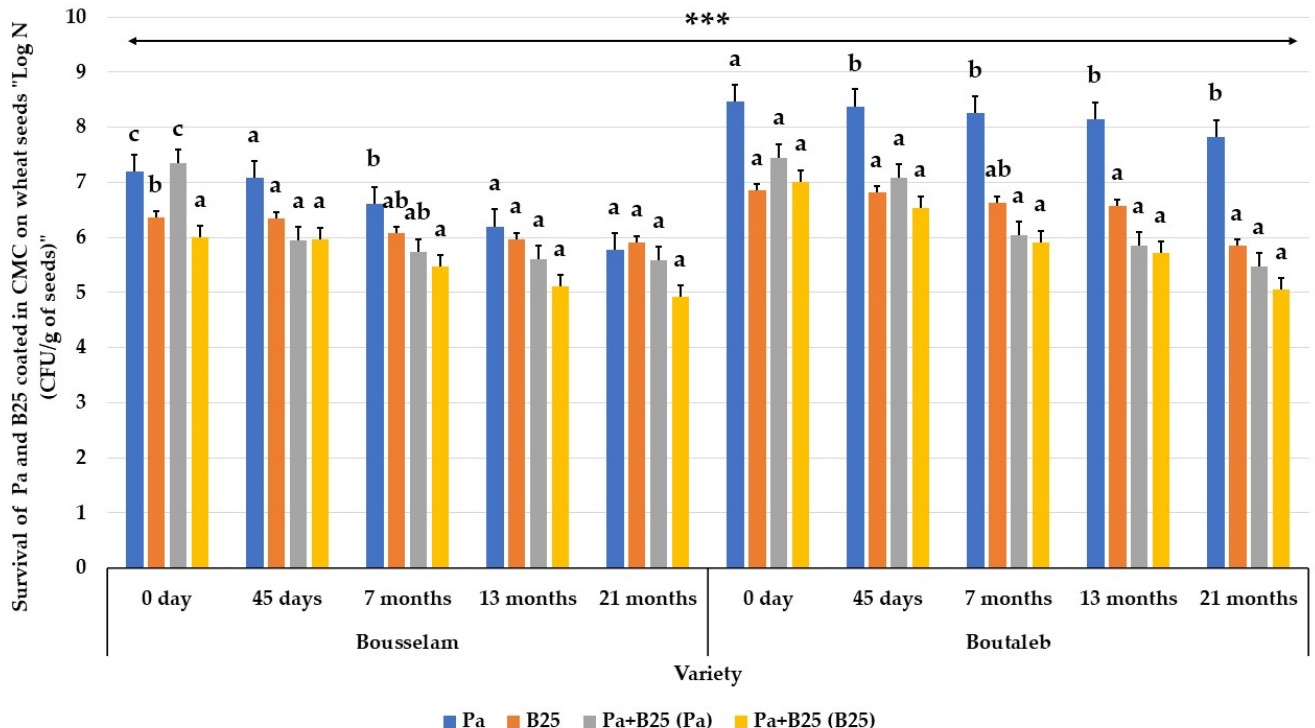

**Figure 15.** Survival of bacteria, Pa and B25 strains (Log N(CFU/g)), in wheat seeds coated with CMC priming with Pa, B25, and Pa + B25 at different period (0, 45 days and 7, 13, and 21 months) of storage. The bar plots represent the mean $\pm$ standard error of three different experiments. IBM SPSS Statistics v.24 was used to perform statistical analysis, using a two-way ANOVA and Tukey's multiple comparison post-test. Significant differences are displayed as: *** $p < 0.001$.

Concerning strain B25, its bacterial density on Bousselam seeds, at the beginning, was about $2.3 \times 10^6$ CFU/g and $7 \times 10^6$. At 21 months, the B25 strain enumeration counts reached $7.95 \times 10^5$ and $7 \times 10^5$ CFU/g in both varieties. However, in consortium, the B25 strain had an initial density of $1.04 \times 10^6$ on Bousselam variety seeds, and $9.90 \times 10^6$ CFU/g on Boutaleb variety seeds. After 21 months, the number of cells decreased to $8.55 \times 10^4$ and $1.13 \times 10^5$ CFU/g for the two varieties, respectively (Figure 15).

## 4. Discussion

In this study, the effects of PGPR inoculation on improving durum wheat plant growth was determined at different growth stages. Bacterial inoculants can be very promising by increasing beneficial traits and inducing dynamics in the soil microbial community for the proper functioning of agro-ecosystems.

The analysis of the germination of two varieties of durum wheat, Bousselam and Boutaleb, followed after treatment of the seeds by coating with CMC and with the bacteria Pa, B25, and their combination (Pa + B25). The effects of the inoculation did not reveal any significant impact on the germination parameters, except for the vigor indexes of length and dry weight of roots and leaves of the Boutaleb variety, which were significantly

stimulated compared to the control. Similar results showed that the germination percentage was not influenced, but other parameters were affected following inoculation with strains of *Bacillus sphaericus* and *Rhizobium* [45]. The germination percentage was not influenced by the contribution of PGPR due to the fact that the seeds can germinate without any problem since they are not subjected to any stress. Seed bio-priming before the germination phase is a method used to improve seed germination under optimal or stressful conditions [46]. The positive impact of priming was found to be more pronounced under stress than in control samples [47]. Thus, other studies on the treatment of durum wheat seeds with bacterial strains of *Bacillus atropheus* and their combinations revealed a significant improvement in germination parameters under salt stress [14]. Moreover, according to Shweta et al. [48], strains of *Pseudomonas fluorescens*, used as groundnut inoculants, improved germination by up to 15% and 30% under biotic stress. Improved emergence and seedling vigor through PGPR inoculation is a prerequisite for better seedling establishment. Seed vigor and viability are important factors influencing seedling establishment, growth, and crop productivity [49]. The vigor index determines the health of the seedling and subsequently the productivity of the plant. The higher the vigor index, the better the yield of the plant.

In another aspect of the study and in order to test the direct effects of the bacterial strains on the plant growth of durum wheat, the coated seeds were sown in sterile compost under aseptic conditions. Inoculation had a significant influence on the growth and development of seedlings, unlike germination. Indeed, the growth parameters (root and leaf length, leaf and root fresh weight, and leaf dry weight) of the Boutaleb variety inoculated with Pa and B25 strains in mono- and co-cultures were improved. On the contrary, in the Bousselam variety, inoculation with the Pa strain alone or in consortium had a significant positive effect on all the morphological parameters, with the exception of root fresh weight. The immediate response to soil inoculation with PGPR varies widely depending on the bacterium, plant species, soil type, inoculum density, and environmental conditions [50]. PGPR must possess specific characteristics for their use as an effective bio-inoculant. They should be able to survive in the soil, compatible with the crop they are inoculated on, and interact with native soil microflora and abiotic factors [51].

The increase in plant growth under sterile conditions is due to the PGP activities of the two strains, Pa and B25. The use of strains with multiple PGP traits should help increase crop productivity on a sustainable basis [52]. Based on their metabolic activity and functional diversity, PGPR have a beneficial effect on plant growth. They help in the promotion of plant growth through nitrogen fixation, phosphate solubilization, and the production of essential plant hormones such as indole acetic acid (IAA), abscisic acid, cytokinin, etc. [53]. The *Pantoea agglomerans* strain, Pa, produced both siderophores and IAA and was found to be effective in phosphate solubilization, nitrogen fixation and in the production of ACC deaminase, $NH_3$, and HCN. The *Bacillus thuringiensis* strain, B25, was also able to fix nitrogen and produce $NH_3$, ACC deaminase, siderophores, and a small amount of IAA in addition to anti-fungal activity against several phytopathogenic fungi. IAA production has a cascading effect on plant development due to its ability to influence root growth, which in turn affects nutrient uptake and, ultimately, plant productivity [54]. This phytohormone is involved in tissue enlargement, division, differentiation, and plant cell responses to light and gravity [55].

However, the response of genotypes to inoculation is another parameter to consider. The intrinsic production capacity of the strain is under the influence of environmental factors [56]. A significant improvement in seedling growth by seed inoculation with the Pa strain attributed this effect to the production of bacterial phytohormones. Inoculation increases the fresh and dry weight of shoots by more than 30% and 60%, respectively, after Bousselam and Boutaleb seeds coated with this strain were sown in sterile compost. In a similar study, Waha variety durum wheat seeds, treated with the Pa strain directly without a coating substrate and grown in sand pots, increased the same parameters by 6.66% and 23.25%, respectively [18]. This difference in growth would probably be due not only to the richness of the substrate but also to the positive impact of priming.

Biochemical parameters such as chlorophyll and total sugars were also improved. PGPR inoculation improves chlorophyll content in rice [52] and wheat [57]. While proline and MDA stressors were reduced. Inoculation of sunflowers (*Helianthus annus*) with PGPR showed inhibitory effects on proline (62%), lipid peroxidation (64%), and anti-oxidant enzyme activity (67%) [58]. In addition, wheat (*Triticum aestivum* L.) seedlings inoculated with *Bacillus subtilis* 10-4 were characterized by a decrease in the level of stress-induced proline and MDA accumulation [59]. Similarly, *Pseudomonas putida* MTCC5279 ameliorated water stress in chickpea (*Cicer arietinum*) plants by modulating membrane integrity, osmolyte accumulation (proline, glycine betaine), and ROS scavenging capacity [60]. The effects of *Pantoea alhagi* LTYR-11ZT resulted in increased accumulation of soluble sugars, decreased proline and MDA accumulation, and decreased chlorophyll degradation in wheat leaves under water stress [61].

The enumeration results of the Pa and B25 strains present in the rhizosphere of sterile compost after 75 days of culture showed the survival of more than $10^5$ CFU/g of the two bacteria (either in mono- or in co-culture) in the two wheat varieties. This demonstrated the bacterial support of the plant throughout the period of plant growth. It is evident that the Pa and B25 strains are able to survive in sterilized soil and compete with other micro-organisms in the non-sterile root rhizosphere. The survival of the strains in the rhizosphere for up to 75 days represents an additional advantage, which allows the plants to take maximum advantage of the symbiotic relationship with the host and in the long term. Failure of PGPR in soil is a major problem due to their non-survival and insufficient effects on crop plants [62]. Crop response to applied bio-fertilizers can be very slow and sometimes unsuccessful as the inoculum will take time to concentrate and colonize the roots [51]. Indeed, maintaining sufficient activity of an inoculant population over an extended period after release often represents the main obstacle to the successful use of microbes as PGP agents [50]. A bio-inoculant is absolutely not effective if it reaches a minimum threshold [63]. It is conceivable that the colonization process is orchestrated by bacterial quorum sensing (QS) [64]. Once a beneficial microbial strain has been able to colonize a host plant, it might be able to display a wide range of activities, contributing to plant fitness [50]. The Pa strain was present inside the roots of both wheat varieties and has the ability to colonize the roots and be endophytic. The ubiquity of *P. agglomerans* as a plant-colonizing bacterium suggests that its long-term association with host plants is well established [64]. Quecine et al. [65] demonstrated that *P. agglomerans* 33.1, which was previously isolated from eucalyptus plants, was able to grow in sugarcane seedlings after systemic colonization. Analysis of the root microflora of soybeans and wheat showed the dominance of *Pantoea* spp., *Paraburkholderia* spp., and *Pseudomonas* spp. [66]. Strains of *P. agglomerans*, isolated as dominant endophyte diazotrophs from the seeds of deep-sea rice, have previously been reported as nitrogen-fixing anaerobic bacteria. Histochemical analysis of the hydroponically grown seedling showed that *P. agglomerans* colonized the root surface, root hairs, root cap, lateral root emergence points, root cortex, and stellate region [67]. Several other works have confirmed that the genus *Pantoea* is represented among plant endophytes, such as wheat [18,61,68], rice [69], sugarcane [70], and sweet potatoes [71].

*Bacillus* spp. are among the genera with the greatest potential for survival. Their ability to form spores thus increases the adaptation of strains to commercial formulation and application in the field. *Bacillus* are preferred as the commercially available PGPR due to the stability of inoculant and the ease of storage of the inoculant product [66]. They are used as bio-fertilizers to increase the productivity and sustainability of economic crops [72]. They protect the plant from several stress conditions through ISR, bio-film formation, secretion of lipopeptides, siderophores, and exopolysaccharides. They also act as an effective denitrifying agent in the agro-ecosystem and maintain soil health through environmentally friendly remediation technologies [73].

Finally in a last aspect, the study was carried out on the plant growth of durum wheat in a non-sterile soil, using the same bacterial treatments (Pa, B25, and Pa + B25) but also



testing the strain *B. velezensis* FZB42 [38] as a positive control. All the bacterial treatments significantly increased the morphological and biochemical parameters of the two wheat varieties compared to the un-inoculated control. Leaf and root length, fresh and dry weight, as well as chlorophyll content were improved by inoculation. It has been noticed that the Pa and B25 strains and the Pa + B25 consortium were effective bio-inoculants in improving the morphological parameters of wheat plant growth. Consortium application or single strain of PGPR for seed bacterization showed improvement in wheat plant growth [74]. The synergistic effects of PGPR inoculation led to an increase in the micronutrient content of wheat plants, which could be due to a positive effect enhancing the translocation of micronutrients from soils to plants [75].

The response of wheat to inoculation with Pa and B25 strains was variable between varieties; the difference in response may be a consequence of the specificity of the host plant. Plant interactions with PGPR strongly depend on the plant genotype [76]. Rice responds to ecologically distinct bacteria by altering its content of flavonoids and hydroxycinnamic acid derivatives [77]. Other studies reveal the variation in microbial community structure mainly based on the genotypic nature of the plant species and also on its geographical location [66]. It is now certain that plant genotype plays a crucial role in the assembly and function of rhizospheric microbiomes and in the selection of bacteria with PGP potential [78].

Improved plant growth in inoculated plants may be a result of the direct PGP activities of the strains used, as it may result from the stimulation of the beneficial rhizospheric bacterial community by PGPR strain inoculation [32].

The difference in the density of culturable bacteria in inoculated and non-inoculated soils was not significant. The density of the microbial community reflects a greater ecological stability of the rhizosphere [79]. These results were comparable to those obtained by Chaudhary et al. [80], who found that soil microbial community density was not significantly affected by bacterial inoculation. This is due to the structure of the rhizobacterial community, which is strongly influenced by the age of the plant. The inoculum of PGPR strains present on the seeds can alter the balance of the bacterial community in the early stages of plant growth [32]. The total bacterial population was high 30 days after planting and decreased with plant age [81].

A positive correlation exists between bacterial community profiles and plant data. An increase in plant yield after inoculation could affect the structure of the bacterial community through an increase in the rate of exudation at higher yields.

In this study, the inoculation increased the number of cultivable endophytic and rhizospheric strains having PGP activities compared to the control in the two wheat varieties. The maximum number of strains having the most PGP activity were isolated from the rhizosphere inoculated with the Pa strain, followed by the rhizosphere inoculated with the consortium Pa + B25 strains, in the two wheat varieties, then inoculated with the B25 and FZB42 strains. Similarly, in the endophytes, the roots of the inoculated plants harbor a number of strains possessing PGP activities greater than the control. Our bio-inoculants may therefore have altered the balance of the bacterial community towards the selection of beneficial populations [32]. The inoculated rhizobacteria likely induced growth hormones and other metabolites that encouraged the proliferation of other native bacteria. Microbial consortia acting in synergistic way have the potential to establish novel microbial communities in the rhizosphere and may result in new PGP effects [82]. According to Farzana et al. [81], *Klebsiella* inoculation can probably improve root growth and increase the secretion of root exudates. Inoculation with *Azospirillum lipoferum* CRT1 affected the size and taxonomic composition of functional communities involved in nitrogen fixation (nifH) or ACC deamination (acdS) [31]. The structure of the soil bacterial community due to the application of *Bacillus thuringiensis* KNU-07 was significantly changed after six weeks post inoculation and increased the growth of pepper plants [83].

Thus, different plant species or genotypes can recruit specific microbiota through differences in root morphology and root exudation patterns [21]. In addition, the composition of root exudates and the structure of the root-associated microbial community are strongly

affected by the growth stage of the plant. Early colonization of the rhizosphere could induce large differences in the structure of the rhizosphere community, and therefore affect plant growth [84]. This process, called the priority effect [85], is due to the advantages that early colonizers often have because they can use space and resources earlier than other micro-organisms and/or because they can produce physical barriers and/or antibiotics that slow the colonization of the plant by subsequent micro-organisms [86]. Functional microbiota can be inherited vertically through seeds [87], or horizontally through the environment. Bacterial diversification results from their ability to perform lateral gene transfer between disparate phylogenetic groups [88]. The growing number of microbial culture libraries can proliferate rapidly and have high mutation rates [89,90]. Individual microbes of the same species could potentially carry different genetic endowments and therefore different functional characteristics [91].

Exploring formulations that provide high cell densities of microbial inoculants and survival rates during storage is therefore a crucial step towards producing successful inoculants [92]. The viability of CMC-coated Pa and B25 strains on wheat seeds was tested for a shelf life of 21 months at room temperature. The initial bacterial load of Pa and B25 strains was more than $10^7$ and $10^6$, respectively. After 45 days, the bacterial biomass remains almost the same, especially in monoculture. A slight decrease to $10^6$ in the monoculture and $10^5$ in the co-culture was observed after 7 and 13 months of storage. The number of viable cells in the formulations ranged from $8.55 \times 10^4$ to $6.7 \times 10^7$ CFU/g seed for 21 months, with a decline of 1 to 2 log10.

Similar results were observed during storage of the *P. ananatis* and *P. fluorescens* co-culture and showed sustained viability—counts decreased slightly but remained around $10^6$ after 55 and 70 days of storage [54]. The bacteria showed good survival capacity during storage. *Bacillus amyloliquefaciens* and *B. pumilus* could also survive in sawdust, rice husks, and talcum powder bio-formulations at a density of $7.0 \times$ log10 CFU/mL for up to 9 months [93].

Individually coated bacterial strains were more stable and survived better compared to co-cultures. These results are in contrast with those from Anwar et al. [54], which showed that the bio-formulation of the co-culture CPP-2 (*P. ananatis* +*P. fluorescens*) was found to be slightly more stable and viable compared to the individual bacterial strains. These results unambiguously document that the strains Pa and B25 can be efficiently encapsulated on seeds using CMC and can survive on seeds for over a year at room temperature.

## 5. Conclusions

This study clearly showed that the bio-priming inoculation of durum wheat seeds with the bacterial strains Pa and B25, either alone or in combination, had significant positive effects on the plant growth parameters of durum wheat. These bio-inoculants significantly improved seed vigor, bio-mass, and leaf and root elongation in both wheat varieties. In addition, bio-formulation with CMC conferred the exceptional survival capacity of these bacteria to the seeds and into the rhizosphere of sterile compost. On the other hand, in the soil, these PGPR affected the microbiome by increasing the beneficial bacterial community of wheat. This suggests the need to better understand the mechanisms of the underlying plant growth promotion induced by these PGPR in the rhizosphere. Thus, the three-way interactions between PGPR inoculums, the native rhizosphere microbiome, and plant roots need to be studied in an integrative manner to understand the plant-growth-promoting process and facilitate the application of these PGPR as a reliable component for managing sustainable agricultural systems.

**Author Contributions:** Conceptualization, N.S., A.S., H.C.-S. and L.B.; methodology, N.S., A.S., H.C.-S., L.L., S.B. and L.B.; software, A.C.B.; validation, N.S., A.S., H.C.-S. and L.B.; formal analysis, N.S., A.S., H.C.-S., A.C.B. and L.B.; investigation, N.S., A.S., H.C.-S., A.C.B., F.N.A., L.L. and L.B.; resources, F.N.A. and L.B.; data curation, N.S., S.B. and A.C.B.; writing—original draft preparation, N.S., A.S. and H.C.-S.; writing—review and editing, A.C.B., L.B. and H.C.-S.; visualization, L.L., A.S., H.C.-S. and L.B.; supervision, A.S., H.C.-S. and L.B.; project administration, A.S., H.C.-S. and L.B.;

funding acquisition, L.B. and F.N.A. All authors have read and agreed to the published version of the manuscript.

**Funding:** This research received no external funding.

**Institutional Review Board Statement:** Not applicable.

**Informed Consent Statement:** Not applicable.

**Data Availability Statement:** Not applicable.

**Conflicts of Interest:** The authors declare no conflict of interest.

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
