# Peer review of "Semi-Arid-Habitat-Adapted Plant-Growth-Promoting Rhizobacteria Allows Efficient Wheat Growth Promotion"

_agronomy, doi:10.3390/agronomy12092221_

Round 1

Reviewer 1 Report

The manuscript entitled “Semi-arid habitat adapted plant growth-promoting rhizobacteria allows efficient wheat growth promotion” by Saadaoui et al. demonstrated the impact of PGPR, i.e. Pantoea agglomerans strain Pa and Bacillus thuringiensis strain B25 of two wheat varieties. As per present findings, the applied PGPR strains after seed priming can be beneficial for improving soil properties and enhancing plant growth and development of durum wheat in favorable and unfavorable environmental conditions. The experiment is well designed and presented very well, and the results are significant. The MS can be acceptable for publication in its current form after minor grammatical errors check the entire MS. 

Author Response

Corresponding Author : The authors thank the reviewers for their comments and appreciate the efforts and the time spending reading and reviewing our manuscript

Reviewer 1

The manuscript entitled “Semi-arid habitat adapted plant growth-promoting rhizobacteria allows efficient wheat growth promotion” by Saadaoui et al. demonstrated the impact of PGPR, i.e. Pantoea agglomerans strain Pa and Bacillus thuringiensis strain B25 of two wheat varieties. As per present findings, the applied PGPR strains after seed priming can be beneficial for improving soil properties and enhancing plant growth and development of durum wheat in favorable and unfavorable environmental conditions. The experiment is well designed and presented very well, and the results are significant. The MS can be acceptable for publication in its current form after minor grammatical errors check the entire MS.

Corresponding Author: The authors thank the reviewer very much for having appreciated this work. 

Reviewer 2 Report

This study clearly showed that bio-priming inoculation of durum wheat seeds with bacterial strains Pa and B25 or in combination had significant positive effects on plant growth parameters of durum wheat. These bio-inoculants significantly improved seed vigor, biomass, leaf and root elongation in both wheat varieties. In addition, bioformulation with CMC conferred an exceptional survival capacity of these bacteria on seeds and in the rhizosphere of sterile compost. On the other hand, in the soil, these PGPR affected the microbiome by increasing the beneficial bacterial community of wheat. This suggests the need to better understand the mechanisms of the underlying plant growth promotion induced by these PGPR in the rhizosphere. Thus, the three-way interactions between PGPR inoculums, the native rhizosphere microbiome, and plant roots need to be studied in an integrative manner to understand the plant growth-promoting process and facilitate the application of these PGPR as a reliable component for managing sustainable agricultural systems.

This work proposes a view that semi-arid habitat adapted plant growth-promoting rhizobacteria allows efficient wheat growth promotion. As such, the matter is of interest, however the paper suffers for four serious limits:

1.     The introduction part needs to be arranged more orderly. First, it introduces the research background and significance of this study. Second, the research progress related to this study is introduced. Third, introduce the important new findings or outstanding contributions of this research.

2.     The plant material part should explain the biological characteristics of the two durum wheat varieties.

3.     Illustrations in the text are best placed after each important result.

4.     The legend of Figure 7B does not match the color of the histogram.

Once the above concerns are fully addressed, the manuscript could be accepted for publication in this journal.

Author Response

Reviewer 2

This work proposes a view that semi-arid habitat adapted plant growth-promoting rhizobacteria allows efficient wheat growth promotion. As such, the matter is of interest, however the paper suffers for four serious limits:

Corresponding Author: The authors appreciate and thank the reviewer for this comments and relevant remarks.

  1. The introduction part needs to be arranged more orderly. First, it introduces the research background and significance of this study. Second, the research progress related to this study is introduced. Third, introduce the important new findings or outstanding contributions of this research.

Corresponding Author: The introduction has been modified.

  1. The plant material part should explain the biological characteristics of the two durum wheat varieties.

Corresponding Author: The characteristics of the two durum wheat varieties were added in “plant material” section.

  1. Illustrations in the text are best placed after each important result.

Corresponding Author: The illustrations (fig 2, 4, and 9) were placed after each figure showing the morphological parameters of plant growth.

  1. The legend of Figure 7B does not match the color of the histogram.

Corresponding Author: Amounts of B25 (orange color) and Pa+B25 (B25)(yellow color) in Figure 7B is zero. Therefore we cannot see theses colors in the figure 7(B).

Reviewer 3 Report

The authors report an analysis on the effects of microbial consortia on the germination and growth of two wheat cultivars. The authors use microbial consortia (two PGPRs) that allow for better absorption of nutrients, water management and defense against pathogens in wheat crops. In particular, an improvement was observed in photosynthetic yield, biomass, and leaf and root formation. Furthermore, the use of this consortium of bacteria has improved the microbial ecosystem of wheat.

The manuscript is clear, and well written. The authors have provided numerous data to support the use of their microbial consortium. However, I believe it is important to highlight the limitations of the use of a few microorganisms. In fact, they only use two PGPRs, however today attention is increasing in the use of more microbial families. This aspect should be emphasized, at least in the discussion (see Tabacchioni S et al. . Identification of Beneficial Microbial Consortia and Bioactive Compounds with Potential as Plant Biostimulants for a Sustainable Agriculture. Microorganisms. 2021; 9(2):426. https://doi.org/10.3390/microorganisms9020426)

Author Response

Reviewer 3

The manuscript is clear, and well written. The authors have provided numerous data to support the use of their microbial consortium. However, I believe it is important to highlight the limitations of the use of a few microorganisms. In fact, they only use two PGPRs, however today attention is increasing in the use of more microbial families. This aspect should be emphasized, at least in the discussion (see Tabacchioni S et al. . Identification of Beneficial Microbial Consortia and Bioactive Compounds with Potential as Plant Biostimulants for a Sustainable Agriculture. Microorganisms. 2021; 9(2):426. https://doi.org/10.3390/microorganisms9020426)

Corresponding Author: The authors thank a lot the reviewer for this suggestion, it is an interesting paper and the reference was added.  

With Best Regards

Dr. Lassaad Belbahri